# Fitting trees to $\ell_1$-hyperbolic distances

**Joon-Hyeok Yim**     Anna C. Gilbert

## Abstract

Building trees to represent or to fit distances is a critical component of phylogenetic analysis, metric embeddings, approximation algorithms, geometric graph neural nets, and the analysis of hierarchical data. Much of the previous algorithmic work, however, has focused on generic metric spaces (i.e., those with no *a priori* constraints). Leveraging several ideas from the mathematical analysis of hyperbolic geometry and geometric group theory, we study the tree fitting problem as finding the relation between the hyperbolicity (ultrametricity) vector and the error of tree (ultrametric) embedding. That is, we define a vector of hyperbolicity (ultrametric) values over all triples of points and compare the $\ell_p$ norms of this vector with the $\ell_q$ norm of the distortion of the best tree fit to the distances. This formulation allows us to define the average hyperbolicity (ultrametricity) in terms of a normalized $\ell_1$ norm of the hyperbolicity vector. Furthermore, we can interpret the classical tree fitting result of Gromov as a $p = q = \infty$ result. We present an algorithm HCCROOTEDTREEFIT such that the $\ell_1$ error of the output embedding is analytically bounded in terms of the $\ell_1$ norm of the hyperbolicity vector (i.e., $p = q = 1$) and that this result is tight. Furthermore, this algorithm has significantly different theoretical and empirical performance as compared to Gromov's result and related algorithms. Finally, we show using HCCROOTEDTREEFIT and related tree fitting algorithms, that supposedly standard data sets for hierarchical data analysis and geometric graph neural networks have radically different tree fits than those of synthetic, truly tree-like data sets, suggesting that a much more refined analysis of these standard data sets is called for.

## 1 Introduction

Constructing trees or ultrametrics to fit given data are both problems of great interest in scientific applications (e.g., phylogeny), algorithmic applications (optimal transport and Wasserstein distances are easier, for example, to compute quickly on trees), data visualization and analysis, and geometric machine learning. An ultrametric space is one in which the usual triangle inequality has been strengthened to $d(x,y) \leq \max\{d(x,z), d(y,z)\}$. A hyperbolic metric space is one in which the metric relations amongst any four points are the same as they would be in a tree, up to the additive constant $\delta$. More generally, any finite subset of a hyperbolic space "looks like" a finite tree.

There has been a concerted effort to solve both of these problems in the algorithmic and machine learning communities, including [1]–[5] among many others. Indeed, the motivation for embedding into hyperbolic space or into trees was at the heart of the recent explosion in geometric graph neural networks [6].

As an optimization problem, finding the tree metric that minimizes the $\ell_p$ norm of the difference between the original distances and those on the tree (i.e., the distortion) is known to be NP-hard for most formulations. [7] showed that it is APX-hard under the $\ell_\infty$ norm. The first positive result also came from [7], which provided a 3-approximation algorithm and introduced a reduction technique from a tree metric to an ultrametric (a now widely used technique). The current best known result is an $O((\log n \log \log n)^{1/p})$ approximation for $1 < p < \infty$ [2], and $O(1)$ approximation for $p = 1$,

37th Conference on Neural Information Processing Systems (NeurIPS 2023).

| Hyperbolicity measure | Value | Distortion |
|---|---|---|
| Gromov's $\delta$-hyperbolicity | $\delta = \|\mathbf{\Delta}_x\|_\infty$ | $\|d - d_T\|_\infty = O(\delta \log n) = O(\|\mathbf{\Delta}_x\|_\infty \log n)$ |
| Average hyperbolicity | $\delta = \frac{1}{\binom{n-1}{3}}\|\mathbf{\Delta}_x\|_1$ | $\|d - d_T\|_1 = O(\delta n^3) = O(\|\mathbf{\Delta}_x\|_1)$ |

Table 1: Connection between $\delta$-hyperbolicity and average hyperbolicity and how these quantities determine the distortion of the resulting tree metric.

recently shown by [1]. Both papers exploit the hierarchical correlation clustering reduction method and its LP relaxation to derive an approximation algorithm.

While these approximation results are fruitful, they are not so practical (the LP result uses an enormous number of variables and constraints). On the more practical side, [8] provided a robust method, commonly known as *Neighbor Join* (which can be computed in $O(n^3)$ time), for constructing a tree. Recently, [9] proposed an $O(n^2)$ method known as *TreeRep* for constructing a tree. Unfortunately, neither algorithm provides a guaranteed bound on the distortion.

The main drawback with all of these results is that they assume almost nothing about the underlying discrete point set, when, in fact, many real application data sets are close to hierarchical or nearly so. After all, why fit a tree to generic data only to get a bad approximation? In fact, perhaps with some geometric assumptions on our data set, we can fit a better tree metric or ultrametric, perhaps even more efficiently than for a general data set.

Motivated by both Gromov's $\delta$-hyperbolicity [4] and the work of Chatterjee and Slonam [10] on average hyperbolicity, we define proxy measures of how *tree-like* a data set is. We note that [4], [11] provide a simple algorithm and analysis to find a tree approximation for which the maximum distortion ($\ell_\infty$ norm) is bounded by $O(\delta \log n)$, where $\delta$ is the hyperbolicity constant. Moreover, this bound turns out to be the best order we can have. In this paper, we go beyond the simple notion of $\delta$-hyperbolicity to define a vector of hyperbolicity values $\mathbf{\Delta}(d)$ for a set of distance values. The various $\ell_p$ norms of this vector capture how tree-like a data set is. Then, we show that the $\ell_q$ norm of the distortion of the best fit tree and ultrametrics can be bounded in terms of this tree proxy. Thus, we give a new perspective on the tree fitting problem, use the geometric nature of the data set, and arrive at, hopefully, better and more efficient tree representations. The table below captures the relationship between the different hyperbolicity measures and the tree fit distortion. We note the striking symmetry in the tradeoffs.

Our main theoretical result can be summarized as

There is an algorithm which runs in time $O(n^3 \log n)$ which returns a tree metric $d_T$ with distortion bounded by the average (or $\ell_1$) hyperbolicity of the distances; i.e.,

$$\|d - d_T\|_1 \leq 8\|\mathbf{\Delta}_x(d)\|_1 \leq 8\binom{n-1}{3} \mathrm{AvgHyp}(d).$$

Additionally, the performance of HCCROTTEDTREEFIT and other standard tree fitting algorithms on commonly used data sets (especially in geometric graph neural nets) shows that they are quite far from tree-like and are not well-represented by trees, especially when compared with synthetic data sets. This suggests that we need considerably more refined geometric notions for learning tasks with these data sets.

## 2 Preliminaries

### 2.1 Basic definitions

To set the stage, we work with a finite metric space $(X, d)$ and we set $|X| = n$, the number of triples in $X$ as $\binom{n}{3} = \ell$, and $r = \binom{n}{4}$ the number of quadruples of points in $X$. In somewhat an abuse of notation, we let $\binom{X}{3}$ denote the set of all triples chosen from $X$ (and, similarly, for all quadruples of points). Next, we recall the notion of hyperbolicity, which is defined via the Gromov product [4]. Given a metric space $(X, d)$, the *Gromov product* of two points $x, y \in X$ with respect to a base point

$w \in X$ is defined as

$$gp_w(x, y) := \frac{1}{2}\left(d(x, w) + d(y, w) - d(x, y)\right).$$

We use the definition of the Gromov product on two points with respect to a third to establish the *four point condition* and define

$$fp_w(d; x, y, z) := \max_{\pi \text{ perm}} [\min(gp_w(\pi x, \pi z), gp_w(\pi y, \pi z)) - gp_w(\pi x, \pi y)]$$

where the maximum is taken over all permutations $\pi$ of the labels of the four points. Since $fp_w(d; x, y, z) = fp_x(d; y, z, w) = fp_y(d; x, z, w) = fp_z(d; x, y, w)$, we sometimes denote the four point condition as $fp(d; x, y, z, w)$. Similarly, we define the *three point condition* as

$$tp(d; x, y, z) := \max_{\pi \text{ perm}} [d(\pi x, \pi z) - \max(d(\pi x, \pi y), d(\pi y, \pi z))]$$

which we use to define ultrametricity.

Following a standard definition of Gromov, a metric space $(X, d)$ is said to be $\delta$-hyperbolic with respect to the base point $w \in X$, if for any $x, y, z \in X$, the following holds:

$$gp_w(x, y) \geq \min(gp_w(y, z), gp_w(x, z)) - \delta.$$

We denote $\mathrm{Hyp}(d) = \delta$, the usual hyperbolicity constant (similarly, $\mathrm{UM}(d)$, is the usual ultrametricity constant). We note that this measure of hyperbolicity is a worst case measure and, as such, it may give a distorted sense of the geometry of the space. A graph which consists of a tree and a single cycle, for instance, is quite different from a single cycle alone but with a worst case measure, we will not be able to distinguish between those two spaces.

In order to disambiguate different spaces, we define the *hyperbolicity* vector as the $\ell$-dimensional vector of all four point conditions with respect to $d$:

$$\mathbf{\Delta}_w(d) = [fp(d; x, y, z, w)] \text{ for all } x, y, z \in \binom{X}{3}.$$

Similarly, we define the *ultrametricity* vector as the $\ell$-dimensional vector of all three point conditions with respect to $d$:

$$\mathbf{\Delta}(d) = [tp(d; x, y, z)] \text{ for all } x, y, z \in \binom{X}{3}.$$

We use the hyperbolicity and ultrametricity vectors to express more refined geometric notions.

We define *p-average hyperbolicity* and *p-average ultrametricity*.

$$\mathrm{AvgHyp}_p(d) = \left(\frac{1}{r} \sum_{x,y,z,w \in \binom{X}{4}} fp(d; x, y, z, w)^p\right)^{1/p} \quad \text{and}$$

$$\mathrm{AvgUM}_p(d) = \left(\frac{1}{\ell} \sum_{x,y,z \in \binom{X}{3}} tp(d; x, y, z)^p\right)^{1/p}$$

If $p = 1$, then the notions are simply the *average* (and we will call them so). Also, for clarity, note the usual hyperbolicity and ultrametricity constants $\mathrm{Hyp}(d)$ and $\mathrm{UM}(d)$ are the $p = \infty$ case.

**Proposition 2.1.** We have the simple relations:

(a) $\mathrm{Hyp}(d) = \max_{x \in X} \|\mathbf{\Delta}_x(d)\|_\infty \geq \|\mathbf{\Delta}_x(d)\|_\infty$ for any $x \in X$.

(b) $\mathrm{UM}(d) = \|\mathbf{\Delta}(d)\|_\infty$.

In the discussion of heirarchical correlation clustering in Section 2.4 and in the analysis of our algorithms in Section 3, we construct multiple graphs using the points of $X$ as vertices and derived edges. Of importance to our analysis is the following combinatorial object which consists of the set of bad triangles in a graph (i.e., those triples of vertices in the graph for which exactly two edges, rather than three, are in the edge set). Given a graph $G = (V, E)$, denote $B(G)$, the set of *bad triangles* in $G$, as

$$B(G) := \left\{(x, y, z) \in \binom{V}{3} \mid |\{(x, y), (y, z), (z, x)\} \cap E| = 2\right\}.$$

## 2.2 Problem formulation

First, we formulate the tree fitting problem. Given a finite, discrete metric space $(X, d)$ and the distance $d(x_i, x_j)$ between any two points $x_i, x_j \in X$, find a tree metric $(T, d_T)$ in which the points in $X$ are among the nodes of the tree $T$ and the tree distance $d_T(x_i, x_j)$ is "close" to the original distance $d(x_i, x_j)$.

While there are many choices to measure how close $d_T$ and $d$ are, in this paper, we focus on the $\ell_p$ error; i.e., $\|d_T - d\|_p$, for $1 \leq p \leq \infty$. This definition is itself a shorthand notation for the following. Order the pairs of points $(x_i, x_j), i < j$ lexicographically and write $d$ (overloading the symbol $d$) for the vector of pairwise distances $d(x_i, x_j)$. Then, we seek a tree distance function $d_T$ whose vector of pairwise tree distances is close in $\ell_p$ norm to the original vector of distances. For example, if $p = \infty$, we wish to bound the *maximum* distortion between any pairs on $X$. If $p = 1$, we wish to bound the total error over all pairs. Similarly, we define the ultrametric fitting problem.

We also introduce the *rooted* tree fitting problem. Given a finite, discrete metric space $(X, d)$ and the distance $d(x_i, x_j)$ between any two points $x_i, x_j \in X$, and a distinguished point $w \in X$, find a tree metric $(T, d_T)$ such that $\|d - d_T\|_p$ is small and $d_T(w, x) = d(w, x)$ for all $x \in X$. Although the rooted tree fitting problem has more constraints, previous work (such as [7]) shows that by choosing the base point $w$ appropriately, the optimal error of the rooted tree embedding is bounded by a constant times the optimal error of the tree embedding. Also, the rooted tree fitting problem is closely connected to the ultrametric fitting problem.

Putting these pieces together, we observe that while considerable attention has been paid to the $\ell_q$ tree fitting problem for generic distances with only mild attention paid to the assumptions on the input distances. No one has considered *both* restrictions on the distances and more sophisticated measures of distortion. We define the $\ell_p/\ell_q$ tree (ultrametric) fitting problem as follows.

**Definition 2.2 ($\ell_\mathbf{p}/\ell_\mathbf{q}$ tree (ultrametric) fitting problem).** Given $(X, d)$ with hyperbolicity vector $\mathbf{\Delta}_x(d)$ and $\mathrm{AvgHyp}_q(d)$ (ultrametricity vector $\mathbf{\Delta}(d)$ and $\mathrm{AvgUM}_q(d)$), find the tree metric $(T, d_T)$ (ultrametric $(X, d_U)$) with distortion

$$\|d - d_T\|_p \leq \mathrm{AvgHyp}_q(d) \cdot f(n) \text{ or } \|d - d_U\|_p \leq \mathrm{AvgUM}_q(d) \cdot f(n)$$

for a growth function $f(n)$ that is as small as possible. (Indeed, $f(n)$ might be simply a constant.)

## 2.3 Previous results

Next, we detail Gromov's classic theorem on tree fitting, using our notation above.

**Theorem 2.3.** *[4] Given a $\delta$-hyperbolic metric space $(X, d)$ and a reference point $x \in X$, there exists a tree structure $T$ and its metric $d_T$ such that*

1. *$T$ is $x$-restricted, i.e., $d(x, y) = d_T(x, y)$ for all $y \in X$.*

2. *$\|d - d_T\|_\infty \leq 2\|\mathbf{\Delta}_x(d)\|_\infty \lceil \log_2(n - 2) \rceil$.*

In other words, we can bound the *maximum distortion* of tree metric in terms of the hyperbolicity constant $\mathrm{Hyp}(d) = \delta$ and the size of our input space $X$.

## 2.4 (Hierarchical) Correlation clustering and ultrametric fitting

Several earlier works ([2], [12]) connected the correlation clustering problem to that of tree and ultrametric fitting and, in order to achieve our results, we do the same. In the correlation clustering problem, we are given a graph $G = (V, E)$ whose edges are labeled "+" (similar) or "-" (different) and we seek a clustering of the vertices into two clusters so as to minimize the number of pairs incorrectly classified with respect to the input labeling. In other words, minimize the number of "-" edges within clusters plus the number of "+" edges between clusters. When the graph $G$ is complete, correlation clustering is equivalent to the problem of finding an optimal ultrametric fit under the $\ell_1$ norm when the input distances are restricted to the values of 1 and 2.

Hierarchical correlation clustering is a generalization of correlation clustering that is also implicitly connected to ultrametric and tree fitting (see [1], [2], [12], [13]). In this problem, we are given a set of non-negativeweights and a set of edge sets. We seek a partition of vertices that is both hierarchical and

minimizes the weighted sum of incorrectly classified pairs of vertices. It is a (weighted) combination of correlation clustering problems.

More precisely, given a graph $G = (V, E)$ with $k + 1$ edge sets $G_t = (V, E_t)$, and $k + 1$ weights $\delta_t \geq 0$ for $0 \leq t \leq k$, we seek a *hierarchical* partition $P_t$ that minimizes the $\ell_1$ objective function, $\sum \delta_t |E_t \Delta E(P_t)|$. It is *hierarchical* in that for each $t$, $P_t$ subdivides $P_{t+1}$.

Chowdury, et al. [13] observed that the Single Linkage Hierarchical Clustering algorithm (SLHC) whose output can be modified to produce an ultrametric that is designed to fit a given metric satisfies a similar property to that of Gromov's tree fitting result. In this case, the distortion bound between the ultrametric and the input distances is a function of the ultrametricity of the metric space.

**Theorem 2.4.** *Given* $(X, d)$ *and the output of SLHC in the form of an ultrametric* $d_U$*, we have*

$$\|d - d_U\|_\infty \leq \|\mathbf{\Delta}(d)\|_\infty \lceil \log_2(n - 1) \rceil.$$

## 2.5 Reductions and equivalent bounds

Finally, we articulate precisely how the tree and ultrametric fitting problems are related through the following reductions. We note that the proof of this theorem uses known techniques from [2] and [1] although the specific results are novel. First, an ultrametric fitting algorithm yields a tree fitting algorithm.

**Theorem 2.5.** *Given* $1 \leq p < \infty$ *and* $1 \leq q \leq \infty$*. Suppose we have an ultrametric fitting algorithm such that for any distance function* $d$ *on* $X$ *(with* $|X| = n$*), the output* $d_U$ *satisfies*

$$\|d - d_U\|_p \leq \mathrm{AvgUM}_q(d) \cdot f(n) \text{ for some growth function } f(n).$$

*Then there exists a tree fitting algorithm (using the above) such that given an input* $d$ *on* $X$ *(with* $|X| = n$*), the output* $d_T$ *satisfies*

$$\|d - d_T\|_p \leq 2 \left( \frac{n - 3}{n} \right)^{1/q} \mathrm{AvgHyp}_q(d) \cdot f(n) \text{ for same growth function } f(n).$$

Conversely, a tree fitting algorithm yields an ultrametric fitting algorithm. From which we conclude that both problems should have the same asymptotic bound, which justifies our problem formulation.

**Theorem 2.6.** *Given* $1 \leq p < \infty$ *and* $1 \leq q \leq \infty$*. Suppose that we have a tree fitting algorithm such that for any distance function* $d$ *on* $X$ *(with* $|X| = n$*), the output* $d_T$ *satisfies*

$$\|d - d_T\|_p \leq \mathrm{AvgHyp}_q(d) \cdot f(n) \text{ for some growth function } f(n).$$

*Then there exists an ultrametric fitting algorithm (using the above) such that given an input* $d$ *on* $X$ *(with* $|X| = n$*), the output* $d_T$ *satisfies*

$$\|d - d_U\|_p \leq 3^{\frac{p-1}{p}} \left( \frac{1}{4} + 2^{q-4} \right)^{1/q} \cdot \mathrm{AvgUM}_q(d) \cdot f(2n) \text{ for same growth function } f(n).$$

Proofs of both theorems can be found in the Appendix 6.1 6.2.

# 3 Tree and ultrametric fitting: Algorithm and analysis

In this section, we present an algorithm for the $p = q = 1$ ultrametric fitting problem. We also present the proof of upper bound and present an example that shows this bound is asymptotically tight (despite our empirical evidence to the contrary). Then, using our reduction from Section 2, we produce a tree fitting result.

## 3.1 HCC Problem

As we detailed in Section 2, correlation clustering is connected with tree and ultrametric fitting and in this section, we present a hierarchical correlation clustering (HCC) algorithm which *bounds* the number of disagreement edges by constant factor of number of bad triangles. We follow with a proposition that shows the connection between bad triangle objectives and the $\ell_1$ ultrametricity vector.

**Definition 3.1** (**HCC with triangle objectives**). Given a vertex set $V$ and $k + 1$ edge sets, set $G_t = (V, E_t)$ for $0 \le t \le k$. The edge sets are hierarchical so that $E_t \subseteq E_{t+1}$ for each $t$. We seek a *hierarchical* partition $P_t$ so that for each $t$, $P_t$ subdivides $P_{t+1}$ and the number of disagreement edges $|E_t \Delta E(P_t)|$ is bounded by $C \cdot |B(G_t)|$ (where $B$ denotes the set of bad triangles) for some constant $C > 0$.

Note that this problem does not include the weight sequence $\{\delta_t\}$, as the desired output will also guarantee an upper bound on $\sum \delta_t |E_t \Delta E(P_t)|$, the usual $\ell_1$ objective.

This proposition relates the $\ell_1$ vector norm of $\mathbf{\Delta}(d)$, the average ultrametricity of the distances, and the bad triangles in the derived graph. This result is why we adapt the hierarchical correlation clustering problem to include triangle objectives.

**Proposition 3.2.** Given distance function $d$ on $\binom{X}{2}$ (with $|X| = n$) and $s > 0$, consider the $s$-neighbor graph $G_s = (X, E_s)$ where $E_s$ denotes $\{(x, y) \in \binom{X}{2} | d(x, y) \le s\}$. Then we have

$$\|\mathbf{\Delta}(d)\|_1 = \ell \cdot \mathrm{AvgUM}(d) = \int_0^\infty |B(G_s)| ds.$$

## 3.2 Main results

Our main contribution is that the HCCTRIANGLE algorithm detailed in Section 3.3 solves our modified HCC problem and its output partition behaves "reasonably" on every level. From this partition, we can construct a good ultrametric fit which we then leverage for a good (rooted) tree fit (using the reduction from Section 2.

**Theorem 3.3.** HCCTRIANGLE *outputs a hierarchical partition* $P_t$ *where* $|E_t \Delta E(P_t)| \le 4 \cdot |B(G_t)|$ *holds for every* $0 \le t \le \binom{n}{2} = k$. *Furthermore, the algorithm runs in time* $O(n^2)$.

By Proposition 3.2 and Theorem 3.3, we can find an ultrametric fit using the HCCTRIANGLE subroutine to cluster our points. This algorithm we refer to as HCCULTRAFIT. Using Theorem 3.3, we have following $\ell_1$ bound.

**Theorem 3.4.** *Given* $d$, HCCULTRAFIT *outputs an ultrametric* $d_U$ *with* $\|d - d_U\|_1 \le 4\|\mathbf{\Delta}\|_1 = 4\ell \cdot \mathrm{AvgUM}_1(d)$. *The algorithm runs in time* $O(n^2 \log n)$.

In other words, HCCULTRAFIT solves the HCC Problem 3.1 with constant $C = 4$. The following proof shows why adapting HCCTRIANGLE as a subroutine is successful.

**Proof of Theorem 3.4 from Theorem 3.3:** Suppose the outputs of HCCTRIANGLE and HCCULTRAFIT are $\{P_t\}$ and $d_U$, respectively. Denote $d_i := d(e_i)$ ($d_0 = 0, d_{k+1} = \infty$) and, for any $s > 0$, set

$$G_s = (X, E_t), \ P_s = P_t \text{ for } d_t \le s < d_{t+1}.$$

Then, for any $x, y \in \binom{X}{2}$, we see that

$$(x, y) \in E_s \Delta E(P_s) \quad \Longleftrightarrow \quad d(x, y) \le s < d_U(x, y) \text{ or } d_U(x, y) \le s < d(x, y).$$

Again, we use the integral notion from Proposition 3.2. Every edge $(x, y)$ will contribute $|E_s \Delta E(P_s)|$ with amount exactly $|d_U(x, y) - d(x, y)|$. Then, by Theorem 3.3,

$$\|d - d_U\|_1 = \int_0^\infty |E_s \Delta E(P_s)| ds$$

$$\le 4 \int_0^\infty |B(G_s)| ds = 4\|\mathbf{\Delta}(d)\|_1,$$

as desired. Assuming that HCCTRIANGLE runs in time $O(n^2)$, HCCULTRAFIT runs in time $O(n^2 \log n)$ as the initial step of sorting over all pairs is needed. Thus ends the proof.

By the reduction argument we discussed in Section 2, we can put all of these pieces together to conclude the following:

**Theorem 3.5.** *Given* $(X, d)$, *we can find two tree fits with the following guarantees:*

- *Given a base point point* $x \in X$, HCCROOTEDTREEFIT *outputs a tree fit* $d_T$ *with* $\|d - d_T\|_1 \le 8\|\mathbf{\Delta}_x(d)\|_1$. *The algorithm runs in* $O(n^2 \log n)$.

- *There exists $x \in X$ where $\|\boldsymbol{\Delta}_x(d)\|_1 \leq \binom{n-1}{3} \mathrm{AvgHyp}_1(d)$. Therefore, given $(X, d)$, one can find a (rooted) tree metric $d_T$ with $\|d - d_T\|_1 \leq 8\binom{n-1}{3} \mathrm{AvgHyp}_1(d)$ in time $O(n^3 \log n)$.*

## 3.3 Algorithm and analysis

---
**Algorithm 1** ISHIGHLYCONNECTED: tests if two clusters are highly connected.

---
  **function** ISHIGHLYCONNECTED
    **Input**: vertex set $X, Y$ and edge set $E$
    For $x \in X$:
      If $|\{y \in Y | (x, y) \in E\}| < \frac{|Y|}{2}$:
        **return** False
    For $y \in Y$:
      If $|\{x \in X | (x, y) \in E\}| < \frac{|X|}{2}$:
        **return** False
    **return** True

---

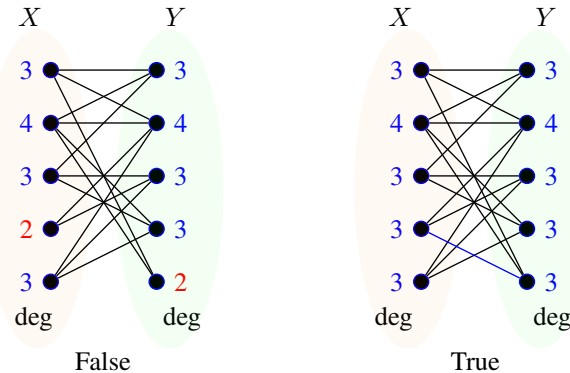

Figure 1: Illustration of highly connectedness condition

---
**Algorithm 2** HCCTRIANGLE: Find HCC which respects triangle objectives

---
  **function** HCCTRIANGLE
    **Input**: $V = \{v_1, \cdots, v_n\}$ and an ordering of all pairs $\{e_1, e_2, \cdots, e_{\binom{n}{2}}\}$ on $V$ so that $E_t = \{e_1, e_2, \cdots, e_t\}$.
    **Desired Output**: *hierarchical* partition $\{P_t\}$ for $0 \leq t \leq \binom{n}{2} = k$ so that $|E_t \Delta E(P_t)| \leq 4 \cdot |B(G_t)|$ holds for every $t$.
    **Init**: $\mathcal{P} = \{\{v_1\}, \{v_2\}, \cdots, \{v_n\}\}$
    $P_0 \leftarrow \mathcal{P}$
    For $t \in \{1, 2, \cdots, \binom{n}{2}\}$:
      Take $e_t = (x, y)$ with $x \in C_x$ and $y \in C_y$ ($C_x, C_y \in \mathcal{P}$)
      If $C_x \neq C_y$:
        If ISHIGHLYCONNECTED$(C_x, C_y, E_t)$ is true:
          add $C = C_x \cup C_y$ and remove $C_x, C_y$ in $\mathcal{P}$.
      $P_t \leftarrow \mathcal{P}$
    **return** $\{P_t\}$

---

In the rest of this subsection, we provide a sketch of the proof that HCCTRIANGLE provides the desired correlation clustering (the detailed proof can be found in Appendix 6.4). We denote the output of HCCTRIANGLE by $\{P_t\}$ and also assume $E = E_t$ and $P_t = \{C_1, C_2, \cdots, C_k\}$. The algorithm HCCTRIANGLE agglomerates two clusters if they are highly connected, adds the cluster to the partition, and iterates. The key to the ordering of the edges input to HCCTRIANGLE is that they are ordered by increasing distance so that the number of edges that are in "disagreement" in

**Algorithm 3** HCCULTRAFIT: Ultrametric fitting algorithm, uses HCCTRIANGLE

---
**function** HCCULTRAFIT($d$)
    Sort $\binom{X}{2}$ as $\{e_1, \cdots, e_{\binom{n}{2}}\}$ so that $d(e_1) \leq d(e_2) \leq \cdots \leq d(e_{\binom{n}{2}})$
    $\{P_t\} = $ HCCTRIANGLE$(X, \{e_t\})$
    $d_U(x, y) \leftarrow d(e_j)$ where $j = \mathrm{argmin}_t(x, y$ are in same cluster in $P_t)$
    **return** $d_U$

---

the correlation clustering is upper bounded by the number of edges whose distances "violate" a triangle relationship. The proof proceeds in a bottom up fashion. (It is clear that the output partition is naturally hierarchical.) We count the number of bad triangles "within" and "between" clusters, which are lower bounded by the number of disagreement edges "within" and "between" clusters, respectively. The proof uses several combinatorial properties which follow from the highly connected condition. This is the key point of our proof.

**From an ultrametricity fit to a rooted tree fit:** Following a procedure from Cohen-Addad, et al. [1], we can also obtain a tree fit with the $\ell_1/\ell_1$ objective. Note that the algorithm HCCROOTEDTREEFIT takes the generic reduction method from ultrametric fitting algorithm but we have instantiated it with HCCULTRAFIT. For a self-contained proof, see the Appendix 6.3.

**Proof of Theorem 3.5:** This proof can be easily shown simply by applying Theorem 2.5 with $p = q = 1$ and $f(n) = 4\binom{n}{3}$.

---
**Algorithm 4** Find a tree fitting given an ultrametric fitting procedure

---
1: **procedure** HCCROOTEDTREEFIT
2:     **Input**: distance function $d$ on $\binom{X}{2}$ and base point $w \in X$
3:     **Output**: (rooted) tree metric $d_T$ which fits $d$ and $d(x, w) = d_T(x, w)$ for all $x \in X$
4:     $M \leftarrow \max_{x \in X} d(x, w)$
5:     $c_w(x, y) \leftarrow 2M - d(x, w) - d(y, w)$ for all $x, y \in \binom{X}{2}$
6:     $d_U \leftarrow$ HCCULTRAFIT$(d + c_w)$
7:     $d_T \leftarrow d_U - c_w$
8:     **return** $d_T$

---

**Running Time** Although ISHIGHLYCONNECTED is seemingly expensive, there is a way to implement HCCTRIANGLE so that all procedures run in $O(n^2)$ time. Thus, HCCULTRAFIT can be implemented so as to run in $O(n^2 \log n)$ time. The detailed algorithm can be found in Appendix 6.5.

**Asymptotic Tightness** Consider $(d, X)$ with $X = \{x_1, \cdots, x_n\}$ and $d(x_1, x_2) = d(x_1, x_3) = 1$ and 2 otherwise. Then we see that $tp(d; x_1, x_2, x_3) = 1$ and 0 otherwise, so that $\|\Delta\|_p = 1$. One can easily check that for any ultrametric $d_U$, $|\epsilon(x_1, x_2)|^p + |\epsilon(x_1, x_3)|^p + |\epsilon(x_2, x_3)|^p \geq 2^{1-p}$ for $\epsilon := d_U - d$. When $p = 1$, $\|d - d_U\|_1 \geq \|\Delta(d)\|_1 = \binom{n}{3}$ AvgUM$(d)$ holds for any ultrametric $d_U$. While HCCULTRAFIT guarantees $\|d - d_U\|_1 \leq 4\|\Delta(d)\|_1 = 4\binom{n}{3}$ AvgUM$(d)$; this shows that our theoretical bound is asymptotically tight.

Examples demonstrating *how* HCCULTRAFIT works and related discussion can be found in Appendix 6.7.1.

## 4 Experiments

In this section, we run HCCROOTEDTREEFIT on several different type of data sets, those that are standard for geometric graph neural nets and those that are synthetic. We also compare our results with other known algorithms. We conclude that HCCROOTEDTREEFIT (HCC) is optimal when the data sets are close to tree-like and when we measure with respect to distortion in the $\ell_1$ sense and running time. It is, however, suboptimal in terms of the $\ell_\infty$ measure of distortion (as to be expected). We also conclude that purportedly hierarchical data sets do not, in fact, embed into trees with low distortion, suggesting that geometric graph neural nets should be configured with different geometric considerations. Appendix 6.9 contains further details regarding the experiments.

| Data set | C-ELEGAN | CS PHD | CORA | AIRPORT |
|---|---|---|---|---|
| $n$ | 452 | 1025 | 2485 | 3158 |
| Hyp($d$) | 1.5 | 6.5 | 11 | 1 |
| AvgHyp($d$) | 0.13 | 0.51 | 0.36 | 0.18 |
| Bound | 158.0 | 1384.5 | 2376.2 | 1547.1 |
| HCC | $0.90_{\pm 0.19}$ | $2.60_{\pm 1.11}$ | $2.42_{\pm 0.44}$ | $1.09_{\pm 0.15}$ |
| Gromov | $1.14_{\pm 0.04}$ | $2.79_{\pm 0.36}$ | $3.38_{\pm 0.13}$ | $1.56_{\pm 0.08}$ |
| TR | $0.83_{\pm 0.16}$ | $2.55_{\pm 1.34}$ | $2.91_{\pm 0.63}$ | $1.28_{\pm 0.21}$ |
| NJ | **0.30** | **1.35** | **1.06** | **0.49** |

Table 2: Connection between hyperbolicity and average hyperbolicity and how these quantities determine the average distortion ($\|d - d_T\|_1 / \binom{n}{2}$) of the resulting tree metric.

| Data set | C-ELEGAN | CS PHD | CORA | AIRPORT |
|---|---|---|---|---|
| HCC | $4.3_{\pm 0.64}$ | $23.37_{\pm 3.20}$ | $19.30_{\pm 1.11}$ | $7.63_{\pm 0.54}$ |
| Gromov | $3.32_{\pm 0.47}$ | $\mathbf{13.24_{\pm 0.67}}$ | $\mathbf{9.23_{\pm 0.53}}$ | $\mathbf{4.04_{\pm 0.20}}$ |
| TR | $5.90_{\pm 0.72}$ | $21.01_{\pm 3.34}$ | $16.86_{\pm 2.11}$ | $10.00_{\pm 1.02}$ |
| NJ | **2.97** | 16.81 | 13.42 | 4.18 |

Table 3: $\ell_\infty$ error (i.e., max distortion)

## 4.1 Common data sets

We used common unweighted graph data sets which are known to be hyperbolic or close to tree-like and often used in graph neural nets, especially those with geometric considerations. The data sets we used are C-ELEGAN, CS PHD from [14], and CORA, AIRPORT from [15]. (For those which contain multiple components, we chose the largest connected component.) Given an unweighted graph, we computed its shortest-path distance matrix and used that input to obtain a tree metric. We compared these results with the following other tree fitting algorithms TREEREP (TR) [9], NEIGHBORJOIN (NJ) [8], and the classical Gromov algorithm. As TREEREP is a randomized algorithm and HCCROOTEDTREEFIT and Gromov's algorithm depends on the choice of a pivot vertex, we run all of these algorithms 100 times and report the average error with standard deviation. All edges with negative weights have been modified with weight 0, as TREEREP and NEIGHBORJOIN both occasionally produce edges with negative weights. Recall, both TREEREP and NEIGHBORJOIN enjoy *no* theoretical guarantees on distortion.

First, we examine the results in Table 2. We note that although the guaranteed bound (of average distortion), $8\binom{n-1}{3} \text{AvgHyp}(d)/\binom{n}{2}$ is asymptotically tight even in worst case analysis, this bound is quite loose in practice; most fitting algorithms perform much better than that. We also see that the $\ell_1$ error of HCCROOTEDTREEFIT is comparable to that of TREEREP, while NEIGHBORJOIN performs much better than those. It tends to perform better when the graph data set is known to be *more* hyperbolic (or tree-like), *despite no theoretical guarantees*. It is, however, quite slow.

Also, we note from Table 3 that Gromov's algorithm, which solves our $\ell_\infty/\ell_\infty$ hyperbolicity problem tends to return better output in terms of $\ell_\infty$ error. On the other hand, its result on $\ell_1$ error is not as good as the other algorithms. In contrast, our HCCROOTEDTREEFIT performs better on the $\ell_1$ objective, which suggests that our approach to this problem is on target.

| | Data set | C-ELEGAN | CS PHD | CORA | AIRPORT |
|---|---|---|---|---|---|
| $O(n^2 \log n)$ | HCC | $0.648_{\pm 0.013}$ | $3.114_{\pm 0.029}$ | $18.125_{\pm 0.330}$ | $28.821_{\pm 0.345}$ |
| $O(n^2)$ | Gromov | $0.055_{\pm 0.005}$ | $0.296_{\pm 0.004}$ | $2.063_{\pm 0.033}$ | $3.251_{\pm 0.033}$ |
| $O(n^2)$ | TR | $0.068_{\pm 0.009}$ | $0.223_{\pm 0.046}$ | $0.610_{\pm 0.080}$ | $0.764_{\pm 0.151}$ |
| $O(n^3)$ | NJ | 0.336 | 4.659 | 268.45 | 804.67 |

Table 4: Running Time(s). For NJ, we implemented the naive algorithm, which is $O(n^3)$. We note that TREEREP produces a tree and not a set of distances; its running times excluded a step which computes the distance matrix, which takes $O(n^2)$ time.

| Initial Tree | $BT(8,3)$ | $BT(5,4)$ | $BT(3,5)$ | $BT(2,8)$ | DISEASE |
|---|---|---|---|---|---|
| $n$ | 585 | 776 | 364 | 511 | 2665 |
| $\|\mathbf{\Delta}_r\|_1/\binom{n-1}{3}$ | $0.01887_{\pm 0.00378}$ | $0.01876_{\pm 0.00375}$ | $0.00150_{\pm 0.00036}$ | $0.00098_{\pm 0.00020}$ | $0.00013_{\pm 0.00010}$ |
| HCC | $\mathbf{0.00443}_{\pm 0.00098}$ | $0.01538_{\pm 0.00733}$ | $\mathbf{0.04153}_{\pm 0.01111}$ | $0.07426_{\pm 0.02027}$ | $\mathbf{0.00061}_{\pm 0.00058}$ |
| Gromov | $0.18225_{\pm 0.00237}$ | $0.44015_{\pm 0.00248}$ | $0.17085_{\pm 0.00975}$ | $0.16898_{\pm 0.00975}$ | $0.18977_{\pm 0.00196}$ |
| TR | $0.01360_{\pm 0.00346}$ | $\mathbf{0.01103}_{\pm 0.00336}$ | $0.06080_{\pm 0.00874}$ | $0.09550_{\pm 0.01887}$ | $0.00081_{\pm 0.00059}$ |
| NJ | $0.03180_{\pm 0.00767}$ | $0.06092_{\pm 0.01951}$ | $0.04309_{\pm 0.00521}$ | $\mathbf{0.04360}_{\pm 0.00648}$ | $0.00601_{\pm 0.00281}$ |

Table 5: Average $\ell_1$ error when $n_e = 500$

In analyzing the running times in Table 4, we notice that HCCROOTEDTREEFIT runs in truly $O(n^2 \log n)$ time. Also, its dominant part is the subroutine HCCTRIANGLE, which runs in $O(n^2)$.

## 4.2 Synthetic data sets

In order to analyze the performance of our algorithm in a more thorough and rigorous fashion, we generate random synthetic distances with low hyperbolicity. More precisely, we construct a synthetic weighted graph from fixed balanced trees. We use $BT(r,h)$ to indicate a balanced tree with branching factor $r$ and height $h$ and DISEASE from [6], an unweighted tree data set. For each tree, we add edges randomly, until we reach 500 additional edges. Each added edge is given a distance designed empirically to keep the $\delta$-hyperbolicity bounded by a value of 0.2.

Then we measured the $\ell_1$ error of each tree fitting algorithm (and averaged over 50 trials). Note that all rooted tree fitting algorithms use a root node (for balanced trees, we used the apex; for DISEASE, we used node 0).

For these experiments, we see quite different results in Table 5. All of these data sets are truly *tree-like*. Clearly, NEIGHBORJOIN performs considerably worse on these data than on the common data sets above, especially when the input comes from a tree with *high branching factor* (note that the branching factor of DISEASE is recorded as 6.224, which is also high). We also note that Gromov's method behaves much worse than all the other algorithms. This is possibly because Gromov's method is known to produce a *non-stretching* tree fit, while it is a better idea to *stretch* the metric in this case. The theoretical bound is still quite loose, but not as much as with the common data sets.

## 5 Discussion

All of our experiments show that it is critical to quantify how "tree-like" a data set is in order to understand how well different tree fitting algorithms will perform on that data set. In other words, we cannot simply assume that a data set is generic when fitting a tree to it. Furthermore, we develop both a measure of how tree-like a data set is and an algorithm HCCROOTEDTREEFIT that leverages this behavior so as to minimize the appropriate distortion of this fit. The performance of HCCROOTEDTREEFIT and other standard tree fitting algorithms shows that commonly used data sets (especially in geometric graph neural nets) are quite far from tree-like and are not well-represented by trees, especially when compared with synthetic data sets. This suggests that we need considerably more refined geometric notions for learning tasks with these data sets.

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

# 6 Appendix

## 6.1 Proof of Theorem 2.5

We will follow the details from [7] (and [1] to address minor issues). This is, in fact, the generic reduction method from ultrametric fitting.

---

**Algorithm 5** Find a tree fitting given an ultrametric fitting procedure

1: **procedure** ULTRAFIT
2:     **Input**: distance function $d$
3:     **Output**: ultrametric $d_U$ which fits $d$
4: **procedure** ROOTEDTREEFIT
5:     **Input**: distance function $d$ on $\binom{X}{2}$ and base point $w \in X$
6:     **Output**: (rooted) tree metric $d_T$ which fits $d$ and $d(w, x) = d_T(w, x)$ for all $x \in X$
7:     Define $m = \max_{x \in X} d(w, x)$, $c_w(x, y) = 2m - d(w, x) - d(w, y)$, and $\beta_x = 2(m - d(w, x))(x \in X)$
8:     $d_{U'} = \text{ULTRAFIT}(d + c_w)$
9:     Restrict $d_U(x, y) = \min(\max(\beta_x, \beta_y, d_{U'}(x, y)), 2m)$
10:     $d_T = d_U - c_w$
11:     **return** $d_T$

---

**Claim:** For any $x, y \in X$, $|d_U(x, y) - (d + c_w)(x, y)| \leq |d_{U'}(x, y) - (d + c_w)(x, y)|$ holds. In other words, the restriction will reduce the error.

**Proof:** It is enough to check the two cases when $d_U$ differs.

Case 1: $\max(\beta_x, \beta_y) = d_U(x, y) > d_{U'}(x, y)$: without loss of generality, we assume that $d_U(x, y) = \beta_x \geq \beta_y$. We have $(d + c_w)(x, y) = d(x, y) + c_w(x, y) \geq (d(w, y) - d(w, x)) + (2m - d(w, x) - d(w, y)) = 2m - 2d(w, x) = \beta_x$, which shows $(d + c_w)(x, y) \geq d_U(x, y) > d_{U'}(x, y)$. Therefore the claim holds.

Case 2: $2m = d_U(x, y) < d_{U'}(x, y)$: since $(d + c_w)(x, y) = 2m - 2gp_w(x, y) \leq 2m$, we have $(d + c_w)(x, y) \leq d_U(x, y) < d_{U'}(x, y)$. Again, the claim holds.

This completes the proof. $\qquad\qquad\square$

**Claim:** The restriction $d_U$ is also an ultrametric.

**Proof:** We need to prove $d_U(x, y) \leq \max(d_U(x, z), d_U(y, z))$ for all $x, y, z \in X$. As $U'$ is ultrametric by assumption, it is enough to check only if

$$d_U(x, y) > d_{U'}(x, y) \text{ or } \max(d_U(x, z), d_U(y, z)) > \max(d_{U'}(x, z), d_{U'}(y, z)) \text{ holds.}$$

Case 1: $d_U(x, y) > d_{U'}(x, y)$: we have $\max(\beta_x, \beta_y) = d_U(x, y) > d_{U'}(x, y)$. Without loss of generality, we assume that $d_U(x, y) = \beta_x \geq \beta_y$. Then we have $d_U(x, z) \geq \max(\beta_x, \beta_z) \geq \beta_x = d_U(x, y)$, which shows $d_U(x, y) \leq \max(d_U(x, z), d_U(y, z))$.

Case 2: $\max(d_U(x, z), d_U(y, z)) < \max(d_{U'}(x, z), d_{U'}(y, z))$: we have $d_U(x, z) < d_{U'}(x, z)$ or $d_U(y, z) < d_{U'}(y, z)$ holds. In either case, we have the value is clipped by the maximum value $2m$ so that $d_U(x, y) \leq \max(d_U(x, z), d_U(y, z)) = 2m$.

Therefore, we conclude that the stronger triangle inequality still holds, which completes the proof. $\qquad\qquad\square$

**Claim:** By the restriction, $d_T$ satisfies tree metric.

**Proof:** First, we need to verify that $d_T$ is a metric. First, we have

$$d_T(x, y) = d_U(x, y) - c_w(x, y) \geq \max(\beta_x, \beta_y) - c_w(x, y) = \max(\beta_x, \beta_y) - \frac{1}{2}(\beta_x + \beta_y)$$

$$= \frac{|\beta_x - \beta_y|}{2} = |d(w, x) - d(w, y)| \geq 0,$$

so that $d_T$ is non-negative. Next, we will prove the triangle inequality: $d_T(x,z) \leq d_T(x,y) + d_T(y,z)$. Without loss of generality, we assume that $d_U(x,y) \geq d_U(y,z)$. Then we have

$$
\begin{aligned}
d_T(x,z) = d_U(x,z) - c_w(x,z) &\leq \max(d_U(x,y), d_U(y,z)) - c_w(x,z) \\
&= d_U(x,y) - c_w(x,y) + c_w(x,y) - c_w(x,z) \\
&= d_T(x,y) + (d(w,z) - d(w,y)) \\
&\leq d_T(x,y) + |d(w,z) - d(w,y)| \leq d_T(x,y) + d_T(y,z).
\end{aligned}
$$

Therefore, $d_T$ is (non-negative) metric. To show it is a tree metric, we examine the four point condition of $d_T$. For any $x,y,z,t \in X$,

$$
\begin{aligned}
d_T(x,y) + d_T(z,t) &= (d_U(x,y) - c_w(x,y)) + (d_U(z,t) - c_w(z,t)) \\
&= (d_U(x,y) + d_U(z,t)) - (c_w(x,t) + c_w(z,t)) \\
&= (d_U(x,y) + d_U(z,t)) - (4m - d(w,x) - d(w,y) - d(w,z) - d(w,t)),
\end{aligned}
$$

so that any pair sum of $d_T$ differs from $d_U$ by $4m - d(w,x) - d(w,y) - d(w,z) - d(w,t)$. As $d_U$ is ultrametric and thus 0-hyperbolic, $d_T$ is also 0-hyperbolic, as desired. Thus, $d_T$ is a tree metric. $\qquad \square$

Finally, we prove that $\ell_p$ error of $d_T$ is bounded as desired. This can be done by

$$
\begin{aligned}
\|d - d_T\|_p = \|(d + c_w) - (d_T + c_w)\|_p = \|d_U - (d + c_w)\|_p \\
\leq \|d_{U'} - (d + c_w)\|_p \leq \mathrm{AvgUM}_q(d + c_w) \cdot f(n),
\end{aligned}
$$

by assumption. For $1 \leq q < \infty$, we have

$$
\begin{aligned}
\frac{1}{n} \sum_{w \in X} (\mathrm{AvgUM}_q(d + c_w))^q &= \frac{1}{n} \sum_{w \in X} \frac{1}{\binom{n}{3}} \sum_{x,y,z \in \binom{X \setminus \{w\}}{3}} tp(d + c_w; x, y, z)^q \\
&= \frac{n-3}{n^2} \sum_{w \in X} \frac{1}{\binom{n-1}{3}} \sum_{x,y,z \in \binom{X \setminus \{w\}}{3}} 2^q f p_w(d; x, y, z)^q \\
&= 2^q \frac{n-3}{n} \cdot \frac{1}{\binom{n}{4}} \sum_{x,y,z,w \in \binom{X}{4}} f p^q(d; x, y, z, w) \\
&= 2^q \frac{n-3}{n} (\mathrm{AvgHyp}_q(d))^q.
\end{aligned}
$$

Therefore, there exists $w \in X$ so that $\mathrm{AvgUM}_q(d + c_w) \leq 2 \left(\frac{n-3}{n}\right)^{1/q} \mathrm{AvgHyp}_q(d)$. The $q = \infty$ case can be separately checked.

Hence, there exists $w \in X$ where such reduction yields a tree fitting with $\ell_p$ error bounded by $2 \left(\frac{n-3}{n}\right)^{1/q} \mathrm{AvgHyp}_q(d) \cdot f(n)$, as desired.

Note that when we use HCCULTRAFIT, then $\max(\beta_x, \beta_y) \leq d_{U'}(x,y) \leq 2m$ always hold so that the clipping is not necessary.

### 6.2 Proof of Theorem 2.6

Our reduction is based on the technique from [16] and Section 9 of [1]. Given $d$ on $X$, we will construct a distance $d'$ on $Z = X \cup Y$ such that $|X| = |Y| = n$ and

$$
d'(z,w) = \begin{cases} d(z,w) & \text{if } z, w \in X \\ M & \text{if } z \in X, w \in Y \text{ or } z \in Y, w \in X \quad \text{for all } z \neq w \in Z, \\ c & \text{if } z, w \in Y \end{cases}
$$

for $M$ large enough and $c$ small enough. Then first we have

**Claim:** If $1 \leq q < \infty$, then $\mathrm{AvgHyp}_q(d') \leq \left(\frac{1}{4} + 2^{q-4}\right)^{1/q} \cdot \mathrm{AvgUM}_q(d)$. For $q = \infty$, we have $\mathrm{Hyp}(d') \leq 2 \mathrm{UM}(d)$.

**Proof:** It can be checked that if at least two of $x, y, z, w$ is from $Y$, then we immediately have $fp(d'; x, y, z, w) = 0$. Therefore, we get

$$\sum_{x,y,z,w \in \binom{Z}{4}} fp(d'; x, y, z, w)^q = \sum_{x,y,z,w \in \binom{X}{4}} fp(d'; x, y, z, w)^q + \sum_{w \in Y} \sum_{x,y,z \in \binom{X}{3}} fp(d'; x, y, z, w)^q$$

$$= \sum_{x,y,z,w \in \binom{X}{4}} fp(d; x, y, z, w)^q + \sum_{w \in Y} \sum_{x,y,z \in \binom{X}{3}} tp(d; x, y, z)^q$$

$$= \binom{n}{4} \mathrm{AvgHyp}_q(d)^q + n \binom{n}{3} \mathrm{AvgUM}_q(d)^q.$$

To bound $\mathrm{AvgHyp}_q$, we will use the fact that

$$fp(d; x, y, z, w) \le \frac{1}{2}[tp(d; x, y, z) + tp(d; x, y, w) + tp(d; x, z, w) + tp(d; y, z, w)].$$

Therefore,

$$\mathrm{AvgHyp}_q(d)^q = \frac{1}{\binom{n}{4}} \sum_{x,y,z,w \in \binom{X}{4}} fp(d; x, y, z, w)^q$$

$$\le \frac{2^{q-2}}{\binom{n}{4}} \sum_{x,y,z,w \in \binom{X}{4}} [tp(d; x, y, z)^q + tp(d; x, y, w)^q + tp(d; x, z, w)^q + tp(d; y, z, w)^q]$$

$$= \frac{2^{q-2}}{\binom{n}{4}} \sum_{x,y,z \in \binom{X}{3}} (n-3) tp(d; x, y, z)^q = \frac{2^q}{\binom{n}{3}} \sum_{x,y,z \in \binom{X}{3}} tp(d; x, y, z)^q = 2^q \mathrm{AvgUM}_q(d)^q.$$

To sum up, we get

$$\mathrm{AvgHyp}_q(d')^q = \frac{1}{\binom{2n}{4}} \sum_{x,y,z,w \in \binom{Z}{4}} fp(d'; x, y, z, w)^q$$

$$= \frac{\binom{n}{4}}{\binom{2n}{4}} \mathrm{AvgHyp}_q(d)^q + \frac{n\binom{n}{3}}{\binom{2n}{4}} \mathrm{AvgUM}_q(d)^q$$

$$\le \frac{1}{16} \mathrm{AvgHyp}_q(d)^q + \frac{1}{4} \mathrm{AvgUM}_q(d)^q$$

$$\le \frac{2^q}{16} \mathrm{AvgUM}_q(d)^q + \frac{1}{4} \mathrm{AvgUM}_q(d)^q = \left(\frac{1}{4} + 2^{q-4}\right) \mathrm{AvgHyp}_q(d)^q.$$

The $q = \infty$ case can be checked separately. $\qquad \square$

Therefore, by the claim above and the assumption, we have a tree fitting $d_T$ which fits $d'$ with $\|d_T - d'\|_p \le \mathrm{AvgHyp}_q(d') \cdot f(2n)$. Our goal is to construct a reduction to deduce an ultrametric fit $d_U$ on $X$, by utilizing $d_T$. First,

**Claim:** If $M$ and $c$ were large and small enough respectively, then $d_T(x_1, y) + d_T(x_2, y) - d_T(x_1, x_2) \ge 2c$ holds for every $x_1, x_2 \in X$ and $y \in Y$.

**Proof:** Suppose the claim fails. As $d(x_1, y) + d(x_2, y) - d(x_1, x_2) = M + M - d(x_1, x_2)$, it suggests that one of three pair distances should have distortion at least $\frac{1}{3}(2M - 2c - d(x_1, x_2))$. This should not happen if $M - c \gg \mathrm{AvgHyp}_q(d') \cdot f(2n)$. $\qquad \square$

We will interpret the above claim as a structural property. Given a tree $T$ which realizes the tree metric $d_T$, one can find a Steiner node $w$ on $x_1, x_2, y$. Then we have $d_T(w, y) = \frac{1}{2}[d_T(x_1, y) + d_T(x_2, y) - d_T(x_1, x_2)] \ge c$. Since it holds for any $x_1, x_2 \in X$, it suggests that every geodesic segment $[x, y]$ for fixed $y \in Y$ should share their paths at least $c$. This observation allows a following *uniformization* on $Y$.

**Claim:** One can refine a tree metric so that $d_T(y_1, y_2) = c$ for all $y_1, y_2 \in Y$ with $\ell_p$ error non-increasing.

---

**Algorithm 6** Refine a tree fit on $Z = X \cup Y$

---

    **procedure** UNIFORMIZATION ON $Y$
2:    **Input**: tree fit $(T, d_T)$ on $Z = X \cup Y$, $1 \le p < \infty$
      **Output**: refined tree $(T', d_{T'})$ on $Z$
4:    $d_{T'}(x_1, x_2) \leftarrow d_T(x_1, x_2) \quad \forall x_1 \ne x_2 \in X$
      $d_{T'}(y_1, y_2) \leftarrow c \quad \forall y_1 \ne y_2 \in Y$
6:    Find $y_0 = \mathrm{argmin}_{y \in Y} \sum_{x \in X} |d_T(x, y) - M|^p$
      $d_{T'}(x, y) \leftarrow d_T(x, y_0) \quad \forall x \in X, y \in Y$
8:    **return** $d_{T'}$

---

**Proof:** Pick $y_0 = \mathrm{argmin}_{y \in Y} \sum_{x \in X} |d_T(x, y) - d(x, y)|^p$. Then we will keep $X \cup \{y_0\}$ with its tree structure and relocate all other $y \ne y_0 \in Y$. As every geodesic segments will share their paths at least $c$, we will pick a point $z$ with $d_T(y, z) = c/2$ on the segment, and draw edge $(z, y)$ for all $y \in Y$ with length $c/2$. Then $d_T(y_1, y_2) = c$ as desired. Furthermore, as $d_{T'}(x, y) = d_T(x, y_0)$ for any $x \in X$ and $y \in Y$,

$$\|d_{T'} - d\|_p^p$$

$$= \sum_{z,w \in \binom{Z}{2}} |d_{T'}(z, w) - d(z, w)|^p$$

$$= \sum_{x,x' \in \binom{X}{2}} |d_{T'}(x, x') - d(x, x')|^p + \sum_{x \in X, y \in Y} |d_{T'}(x, y) - M|^p + \sum_{y,y' \in \binom{Y}{2}} |d_{T'}(y, y') - c|^p$$

$$\le \sum_{x,x' \in \binom{X}{2}} |d_T(x, x') - d(x, x')|^p + \sum_{x \in X, y \in Y} |d_T(x, y_0) - M|^p + \sum_{y,y' \in \binom{Y}{2}} 0$$

$$= \sum_{x,x' \in \binom{X}{2}} |d_T(x, x') - d(x, x')|^p + \sum_{y \in Y} \sum_{x \in X} |d_T(x, y_0) - M|^p$$

$$\le \sum_{x,x' \in \binom{X}{2}} |d_T(x, x') - d(x, x')|^p + \sum_{y \in Y} \sum_{x \in X} |d_T(x, y) - M|^p \le \|d_T - d\|_p^p,$$

as desired. $\qquad\qquad\qquad\qquad\qquad\qquad\qquad\qquad\qquad\qquad\qquad\qquad\qquad\qquad\qquad$ $\square$

---

**Algorithm 7** Construction of a restricted tree with respect to given root $r$

---

    **procedure** RESTRICTTREE
2:    **Input**: Tree fitting $(T, d_T)$ on $Z$
      **Output**: Tree fitting $d_T$ such that $d_T(x, y) = M$ for all $x \in X$ and $y \in Y$
4:    **for** each $x \in X$:
        **if** $d_T(x, y_0) > M$:
6:        Relocate $x$ to point on geodesic $[x, y_0]$ so that $d_T(x, y_0) = M$
        **else if** $d_T(x, y_0) < M$:
8:        Add edge $(x, x')$ with length $M - d_T(x, y_0)$ and relocate $x$ to $x'$
      **return** $(T, d_T)$

---

**Claim:** One can refine a tree metric so that $d_{T'}(x, y) = M$ for all $x \in X$ and $y \in Y$ with $\ell_p$ error not greater than $3^{(p-1)/p}$ times the original error. Furthermore, if we restrict $d_{T'}$ on $X$, then it is an ultrametric.

**Proof:** We will use the algorithm RESTRICTTREE to obtain the restricted tree as described. It is clear that such procedure will successfully return a tree metric $d_T$ with $d_T(x, y) = M$ for all $x \in X$ and $y \in Y$. Furthermore, as Denote $\epsilon := d_T - d$. Then for any pair $x_1, x_2 \in X$, its distortion does not increase greater than $|\epsilon(x_1, y_0)| + |\epsilon(x_2, y_0)|$, which is the amount of relocation. We can utilize this

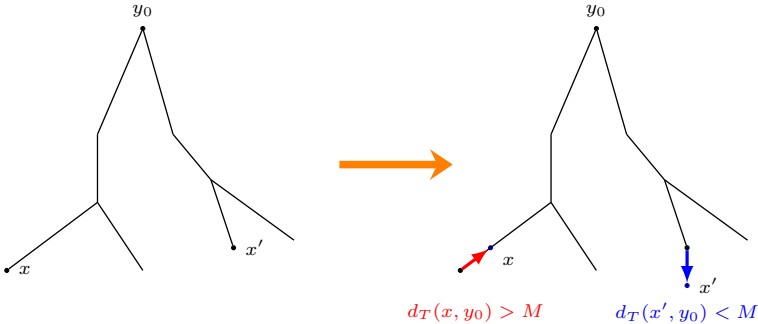

Figure 2: This figure depicts how RESTRICTTREE works. The idea is we can relocate every vertices so that $d(x, y_0) = d_T(x, y_0)$ holds for every $x \in X$.

fact by

$$\|d_{T'} - d\|_p^p = \sum_{z,w \in \binom{Z}{2}} |d_{T'}(z, w) - d(z, w)|^p$$

$$= \sum_{x,x' \in \binom{X}{2}} |d_{T'}(x, x') - d(x, x')|^p + \sum_{x \in X, y \in Y} |d_{T'}(x, y) - M|^p + \sum_{y,y' \in \binom{Y}{2}} |d_{T'}(y, y') - c|^p$$

$$= \sum_{x,x' \in \binom{X}{2}} |d_{T'}(x, x') - d(x, x')|^p$$

$$\leq \sum_{x,x' \in \binom{X}{2}} (|\epsilon(x, y_0)| + |\epsilon(x, x')| + |\epsilon(x', y_0)|)^p$$

$$\leq 3^{p-1} \sum_{x,x' \in \binom{X}{2}} (|\epsilon(x, y_0)|^p + |\epsilon(x, x')|^p + |\epsilon(x', y_0)|^p)$$

$$\leq 3^{p-1} \left( \sum_{x,x' \in \binom{X}{2}} |\epsilon(x, x')|^p + (n-1) \sum_{x \in X} |\epsilon(x, y_0)|^p \right)$$

$$\leq 3^{p-1} \left( \sum_{x,x' \in \binom{X}{2}} |\epsilon(x, x')|^p + n \sum_{x \in X} |\epsilon(x, y_0)|^p \right) = 3^{p-1} \|d_T - d\|_p^p,$$

as desired. Then by the restriction, $d_T(x, y_0) = M$ holds for all $x \in X$. Therefore,

$$tp(d_{T'}; x_1, x_2, x_3) = fp(d_{T'}; x_1, x_2, x_3, y_0) = 0$$

holds for all $x_1, x_2, x_3 \in \binom{X}{3}$, which shows that $d_{T'}$ on $X$ is an ultrametric. $\qquad\square$

Denote the restricted metric as $d_U$ (on $\binom{X}{2}$)). Then we have

$$\|d_U - d\|_p = \|d_{T''} - d'\|_p \leq 3^{\frac{p-1}{p}} \|d_{T'} - d'\|_p \leq 3^{\frac{p-1}{p}} \|d_T - d'\|_p$$

$$\leq 3^{\frac{p-1}{p}} \left( \frac{1}{4} + 2^{q-4} \right)^{1/q} \cdot \mathrm{AvgUM}_q(d) \cdot f(2n),$$

as desired.

## 6.3 Self-contained proof of ultrametric fit to rooted tree fit

Choose any base point $w \in X$ and run HCCROOTEDTREEFIT. Note that $d + c_w = 2(M - gp_w)$. Then by 3.4, we see that the output $d_T$ satisfies

$$\|d - d_T\|_1 \leq 8\|\mathbf{\Delta}(d + c_w)\|_1 = 8\|\mathbf{\Delta}(M - gp_w)\|_1 = 8\|\mathbf{\Delta}_w(d)\|_1.$$

Although $\|\mathbf{\Delta}_w(d)\|_1$ itself does not satisfy the guaranteed bound, we know that

$$\sum_{w \in X} \|\mathbf{\Delta}_w(d)\|_1 = \sum_{w \in X} \sum_{x,y,z \in \binom{X \setminus \{w\}}{3}} fp_w(d; x, y, z)$$

$$= 4 \sum_{x,y,z,w \in \binom{X}{4}} fp(d; x, y, z, w) = 4 \binom{n}{4} \text{AvgHyp}(d)$$

so there exists an output $d_T$ (with appropriately chosen $w$) with its $\ell_1$ error bounded by $8 \cdot \frac{4}{n}\binom{n}{4} \text{AvgHyp}(d) = 8\binom{n-1}{3} \text{AvgHyp}(d)$. As we need to check every base point $w \in X$, this procedure runs in time $O(n^3 \log n)$.

### 6.4   Proof of Theorem 3.3

We begin by recalling the definition of highly connectedness.

**Definition 6.1.** Given a graph $G = (V, E)$ and two disjoint vertex sets $X$ and $Y$, the pair $(X, Y)$ is said to be *highly connected* if both

$$\text{for all } x \in X, \ |\{y \in Y | (x, y) \in E\}| \geq \frac{|Y|}{2}, \text{ and}$$

$$\text{for all } y \in Y, \ |\{x \in X | (y, x) \in E\}| \geq \frac{|X|}{2}$$

hold.

Our first lemma shows that every clusters should contain reasonably many edges "within", so that the number of "false positive" edges can be bounded. Because highly connectedness determines whether at least half of the edges have been connected, we can expect that the degree of every node should be at least half of the size of clusters.

**Lemma 6.2.** For any $C \in P_t$ with $|C| \geq 2$ and $x \in C$, $|\{x' \in C | (x, x') \in E_t\}| \geq \frac{|C|}{2}$.

*Proof.* We use induction on $t$. The lemma is obviously true when $t = 0$. Next, suppose we pick $C \in P_t$ with $|C| \geq 2$ and the cluster $C$ is formed at step $t_0 \leq t$. If $t_0 < t$, since $P_t$ is increasing, it is clear that

$$|\{x' \in C | (x, x') \in E_t\}| \geq |\{x' \in C | (x, x') \in E_{t_0}\}| \geq \frac{|C|}{2} \quad \text{for all } x \in C.$$

Therefore, it is enough to check when $t_0 = t$, i.e., $C$ is the newly added cluster at step $t$.

Suppose $C$ is achieved by merging $C = C_a \cup C_b$. We then have two cases to check, depending on the sizes of $C_a$ and $C_b$.

Case 1: $|C_a| = 1$ or $|C_b| = 1$: Without loss of generality, assume that $|C_a| = 1$ and let $C_a = \{a\}$. If $C_b$ is also a singleton, then there is nothing to prove. If not, then, we must have $(a, y) \in E_t$ for all $y \in C_b$ in order to be *highly connected*. And by the induction hypothesis,

$$|\{y' \in C_b | (y, y') \in E_t\}| \geq |\{y' \in C_b | (y, y') \in E_{t-1}\}| \geq \frac{|C_b|}{2} \quad \text{for all } y \in C_b.$$

Hence, we have

$$|\{y' \in C | (a, y') \in E_t\}| = |C_b| = |C| - 1 \geq \frac{|C|}{2}$$

and for all $y \in C_b$

$$|\{y' \in C | (y, y') \in E_t\}| = |\{y' \in C_b | (y, y') \in E_t\}| + 1 \geq \frac{|C_b|}{2} + 1 > \frac{|C|}{2},$$

which completes the proof of Case 1.

Case 2: $|C_a| > 1$ and $|C_b| > 1$: By the induction hypothesis, we have

$$|\{x' \in C_a | (x, x') \in E_t\}| \geq |\{x' \in C_a | (x, x') \in E_{t-1}\}| \geq \frac{|C_a|}{2} \quad \text{for all } x \in C_a,$$

and similarly for $C_b$. As $C_a$ and $C_b$ are both *highly connected*, we have

$$|\{y' \in C_b | (x, y') \in E_t\}| \geq \frac{|C_b|}{2} \quad \text{for all } x \in C_a,$$

$$|\{x' \in C_b | (y, x') \in E_t\}| \geq \frac{|C_a|}{2} \quad \text{for all } y \in C_b.$$

Therefore, for any $x \in C_a$,

$$|\{z' \in C | (x, z') \in E_t\}| = |\{x' \in C_a | (x, x') \in E_t\}| + |\{y' \in C_b | (x, y') \in E_t\}|$$
$$\geq \frac{|C_a|}{2} + \frac{|C_b|}{2} = \frac{|C|}{2},$$

while each inequality comes from the induction hypothesis and the connectivity condition. We can similarly show the property for any $y \in C_b$, which completes the proof of Case 2.

This completes the proof. $\qquad\square$

Next, we prove the following isoperimetric property:

**Lemma 6.3.** For $C \in P_t$ denote $\partial X := \{(x, y) \in E_t | x \in X, y \in C \setminus X\}$ for $X \subset C$. Then for any proper $X$,

$$|\partial X| \geq \frac{|C|}{2}.$$

*Proof.* Without loss of generality, assume that $1 \leq |X| \leq \frac{|C|}{2}$. Then for any $x \in X$, by 6.2, there are at least $\frac{|C|}{2} - |X| + 1$ edges which connects $x$ and other vertices not in $X$. Therefore, we have

$$|\partial X| \geq |X| \left( \frac{|C|}{2} - |X| + 1 \right) \geq \frac{|C|}{2}.$$

$\qquad\square$

As our version of the HCC problem bounds the number of edges in disagreement in terms of the bad triangles, with the next lemmas, we count the number of bad triangles $|B(G_t)|$ and bound the number of edges in disagreement. Recall that a bad triangle is defined as a triplet of edges in which only two of the three possible edges belong to $E_t$. For each cluster $C \in P_t$, denote the number of bad triangles *inside* $C$ as $T_C$. And for each cluster pair $C, C' \in P_t$, we count *some* of the bad triangles *between* $C$ and $C'$ as $T_{C,C'}$. More precisely,

**Definition 6.4.** Given a partition $P_t$, denote

$$T_C := \{(x, y, z) | x, y, z \in C \text{ and } (x, y, z) \in B(G_t)\} \quad \text{for } C \in P_t,$$

$$T_{(C,C')} := \{(x, x', y) | x, x' \in C, y \in C' \text{ and } (x, x'), (x, y) \in E_t, (x', y) \notin E_t\}$$
$$\cup \{(x, y, y') | x \in C, y, y' \in C' \text{ and } (y, y'), (x, y) \in E_t, (x, y') \notin E_t\} \quad \text{for } (C, C') \in \binom{P_t}{2}.$$

Note that $T_{C,C'}$ does not count *every* bad triangle between $C$ and $C'$, as there might be bad triangles in which the missing edge is *inside* the cluster. With these definitions, we have

$$\sum_{C \in P_t} |T_C| + \sum_{(C,C') \in \binom{P_t}{2}} |T_{(C,C')}| \leq |B(G_t)|.$$

**Proposition 6.5.** For any cluster $C \in P_t$, the number of edges *not* in $C$ is bounded by $|T_C|$. That is,

$$|\{(x, y) \notin E | x, y \in C\}| \leq |T_C|.$$

(In fact, it is bounded by $|T_C|/2$.)

*Proof.* We will prove that for any $e = (x, y) \notin E_t$ with $x, y \in C$, there exist at least two elements in $T_C$ that contain $e$, which implies our result. If $|C| = 1$, then there is nothing to prove. For any $|C| \geq 2$ and $(x, y) \notin E_t$, by Lemma 6.2, there are at least $\frac{|C|}{2}$ neighbors of $x$ and $y$ (we will denote each as $N_x$ and $N_y$, respectively). Since $N_x, N_y \subset C \setminus \{x, y\}$, we have

$$|N_x \cap N_y| = |N_x| + |N_y| - |N_x \cup N_y| \geq \frac{|C|}{2} + \frac{|C|}{2} - (|C| - 2) = 2.$$

For any $z \in N_x \cap N_y$, $(x, y, z) \in T_C$ by definition, which proves the assertion. $\quad\square$

**Lemma 6.6.** For $G = (V, E)$ and two disjoint subsets $X, Y \subset V$, suppose

$$|\{x' \in X | (x, x') \in E\}| \geq \frac{|X|}{2} \text{ for all } x \in X \text{ and } |\{y' \in Y | (y, y') \in E\}| \geq \frac{|Y|}{2} \text{ for all } y \in Y.$$

We will further assume that $X$ and $Y$ are not highly connected. Then we have

$$|\{(x, y) \in E | x \in X, y \in Y\}| \leq 4 \cdot (|\{(x, x', y) | x, x' \in X, y \in Y \text{ and } (x, x'), (x, y) \in E, (x', y) \notin E\}|$$
$$+ |\{(x, y, y') | x \in X, y, y' \in Y \text{ and } (y, y'), (x, y) \in E, (x, y') \notin E\}|)$$

*Proof.* The key argument here is to use Lemma 6.3 and count the boundary sets. Define $N_Y(x) := \{y \in Y | (x, y) \in E\}$ for $x \in X$ (and $N_X(y)$ for $y \in Y$ respectively.) Then for any $(y, y') \in \partial N_Y(x)$, $(x, y, y')$ is a bad triangle and it contributes the right hand side. In other words, the right hand side of the above equation is upper bounded by

$$(\star) = 4 \cdot \left( \sum_{x \in X} |\partial N_Y(x)| + \sum_{y \in Y} |\partial N_X(y)| \right).$$

By 6.3, we know that for $N_Y(x) \neq \emptyset$ or $Y$, $|\partial N_Y(x)| \geq \frac{|Y|}{2}$. We will count such $x \in X$ and $y \in Y$ which its neighbor set is *proper*, in order to make a lower bound. We divide the rest of the analysis into three cases:

Case 1: There exists $x_0 \in X$ such that $N_Y(x_0) = \emptyset$: then for any $y \in Y$, $x_0 \notin N_X(y)$ so that $N_X(y)$ cannot be $X$. Thus, $N_X(y) \neq \emptyset$ immediately leads that $|\partial N_X(y)| \geq \frac{|X|}{2}$. We have

$$|\{(x, y) \in E | x \in X, y \in Y\}| \leq |X| \cdot |\{y \in Y | N_X(y) \neq \emptyset\}|$$
$$\leq \sum_{y \in Y, N_X(y) \neq \emptyset} |X|$$
$$\leq 2 \sum_{y \in Y, N_X(y) \neq \emptyset} |\partial N_X(y)|$$
$$\leq (\star) \leq \text{RHS},$$

which proves the lemma.

Case 2: There exists $y_0 \in Y$ such that $N_X(y_0) = \emptyset$: this case is proven as we did in Case 1.

Case 3: For any $x \in X$ and $y \in Y$, $N_Y(x) \neq \emptyset$ and $N_X(y) \neq \emptyset$, we will use the assumption that $X$ and $Y$ are *not* highly connected. With this assumption, we can also assume without loss of generality that there is an $x_0 \in X$ such that $|N_Y(x_0)| < \frac{|Y|}{2}$. Then for any $y \notin N_Y(x_0)$, $N_X(y) \neq \emptyset$ and $N_X(y) \neq X$ (as $x_0 \notin N_X(y)$). Thus, the boundary set exists and has at

least $\frac{|X|}{2}$ elements. Therefore,

$$\text{RHS} \geq (\star) \geq 4 \sum_{y \in Y} |\partial N_X(y)|$$

$$\geq 4 \sum_{y \in Y, y \notin N_Y(x_0)} |\partial N_X(y)|$$

$$\geq 4 \sum_{y \in Y, y \notin N_Y(x_0)} \frac{|X|}{2},$$

$$= 4 \cdot \frac{|X|}{2} \cdot (|Y| - |N_Y(x_0)|)$$

$$> 4 \cdot \frac{|X|}{2} \cdot \frac{|Y|}{2} = |X| \cdot |Y| \geq \text{LHS}.$$

This completes the proof. $\qquad\square$

**Proposition 6.7.** For any cluster pair $(C, C') \in P_t$, the number of edges *between* $C$ and $C'$ is bounded by $4|T_{(C,C')}|$. (i.e., $|\{(x, y) \in E | x \in C, y \in C'\}| \leq 4|T_{(C,C')}|$.)

*Proof.* Here, we will use an induction on $t$. When $t = 0$, then there is nothing to prove. Suppose the induction hypothesis holds for $t_0 < t$.

Case 1: $e_t$ does not connect two vertices between $C$ and $C'$ and $C, C'$ are not added at step $t$: then by induction hypothesis, the proposition holds at step $t - 1$, which is also true at step $t$ (As the left hand side is invariant and $|T_{(C,C')}|$ is increasing).

Case 2: $e_t = (x, y)$ for $x \in C$ and $y \in C'$: then it follows that our algorithm decided not to merge two clusters, which means $C$ and $C'$ are *not* highly connected. Then by Lemma 6.6, the number of edges is bounded by $4|T_{(C,C')}|$ as desired.

Case 3: $e_t$ does not connect two vertices between $C$ and $C'$, but $C$ or $C'$ is newly formed at step $t$: note that both clusters cannot be generated at the same time, so we assume without loss of generality that $C$ is the new cluster and assume it is generated by $C = C_a \cup C_b$. Then by the induction hypothesis, we have

$$\{(x, y) \in E | x \in C, y \in C'\}| = \{(x, y) \in E | x \in C_a, y \in C'\}| + \{(x, y) \in E | x \in C_b, y \in C'\}|$$

$$\leq 4(|T_{(C_a, C')}| + |T_{(C_b, C')}|) \quad \text{(by the induction hypothesis)}$$

$$\leq 4|T_{(C,C')}|.$$

This completes the proof. $\qquad\square$

We are now ready to prove Theorem 3.3.

**Proof of Theorem 3.3:** For $P_t = \{C_1, C_2, \cdots, C_k\}$, denote

$$e_i := |\{(x, x') \notin E | x, x' \in C_i\}|,$$

$$e_{i,j} := |\{(x, y) \in E | x \in C_i, y \in C_j\}|.$$

Then by the above propositions, we have

$$|E_t \Delta E(P_t)| = \sum_i e_i + \sum_{i<j} e_{i,j}$$

$$\leq \sum_i T_{C_i} + \sum_{i<j} 4T_{(C_i, C_j)}$$

$$\leq 4 \left( \sum_i T_{C_i} + \sum_{i<j} T_{(C_i, C_j)} \right) \leq 4 \cdot |B(G_t)|.$$

## 6.5 Implementation details

To detail our implementation, we will introduce the following variables.

For $x \in X$ and cluster $C$, denote $D(x, C) := |\{y \in C | (x, y) \in E\}|$. We further denote the *connectivity condition* $H(x, C)$ as IF($D(x, C) \geq \frac{|C|}{2}$).
Also for given two clusters $C_a$ and $C_b$, we will denote $M(C_a, C_b) := |\{x \in C_a | H(x, C_b) = 1\}|$.
We can easily check that two disjoint clusters $C_a$ and $C_b$ are highly connected if and only if

$$M(C_a, C_b) + M(C_b, C_a) = |C_a| + |C_b|.$$

---

**Algorithm 8** HCCTRIANGLE: Actual Implementation

---

  **function** HCCTRIANGLE
    **Input**: $X = \{x_1, \cdots, x_n\}, \{e_t\}$
    Initialize $D, H, M$ as 0 for vertex set $X$
    Initialize $\mathcal{P} = \{\{x_1\}, \{x_2\}, \cdots, \{x_n\}\}, \mathcal{E} = \emptyset$
    For $t \in \{1, 2, \cdots, \binom{n}{2}\}$:
      Take $e_t = (x, y)$ with $x \in C_a$ and $y \in C_b$ ($C_a, C_b \in \mathcal{P}$)
      Add $D(x, C_b), D(y, C_a)$ by 1
      Update $H(x, C_b)$ and $H(y, C_a)$
      If $H$ has been updated, update $M(C_a, C_b)$ and $M(C_b, C_a)$ accordingly
      If $C_a \neq C_b$ and $M(C_a, C_b) + M(C_b, C_a) = |C_a| + |C_b|$:
        Remove $C_a, C_b$ and add $C_{new} := C_a \cup C_b$ in $\mathcal{P}$
        $D(:, C_{new}) \leftarrow D(:, C_a) + D(:, C_b)$
        Compute $H(:, C_{new})$ and $M(:, C_{new})$ (by using $D(:, C_{new})$)
        $M(C_{new}, :) \leftarrow M(C_a, :) + M(C_b, :)$
      $P_t \leftarrow \mathcal{P}$
    **return** $\{P_t\}$

---

One can observe that an iteration only takes $O(1)$ when $C_a$ and $C_b$ are not merged. When we merge $C_{new}$, we need time $O(n)$ to compute new information of $C_{new}$. Since such events occur exactly $n - 1$ times, we can conclude that HCCTRIANGLE overall runs in time $O(n^2)$.

## 6.6 Asymptotic tightness of Gromov's distortion bound

While it is well-known that Gromov's result is asymptotically tight, we give a self-contained sketch of the proof and provide an example that witnesses the bound. We detail the construction given by [17]. Consider the Poincaré disk $\mathbb{H}^2$ and its hyperbolic $n = 4m$-gon $H_{n,r}$ as a set of equally spaced $n$ points around the circle of radius $r$ in $\mathbb{H}^2$. Label all the points as $x_1, x_2, \ldots, x_{4m}$, in the counterclockwise order, and define $y = x_{3m}$. Suppose $d_T$ is a tree metric fit to $d$ and denote the Gromov product with respect to $y$ as $gp_y$. Then

$$gp_y(x_0, x_{2m}) \geq \min(gp_y(x_0, x_1), gp_y(x_1, x_{2m})) \geq \min(gp_y(x_0, x_1), \min(gp_y(x_1, x_2), gp_y(x_2, x_{2m})))$$
$$\geq \quad \cdots$$
$$\geq \min_{i \in \{0, 1, \ldots, 2m-1\}} gp_y(x_i, x_{i+1}).$$

Therefore there exists an $i$ such that

$$d_T(x_i, y) + d_T(x_{i+1}, y) - d_T(x_i, x_{i+1}) \leq d_T(x_0, y) + d_T(x_{2m}, y) - d_T(x_0, x_{2m}).$$

On the other hand, for $r, n$ large enough,

$$[d(x_i, y) + d(x_{i+1}, y) - d(x_i, x_{i+1})] - [d(x_0, y) + d(x_{2m}, y) - d(x_0, x_{2m})] \approx 2\log(\sin(\pi/m)) \approx 2\log n,$$

shows one of six pair distances should have distortion at least $\approx 1/3 \log n = \Omega(\log n)$.

## 6.7 Examples

### 6.7.1 Example for **HCCULTRAFIT**

For the following distances $d$ on $X = \{a, b, c, d, e, f\}$, we will compute the ultrametric fit using HCC. We first sort all the pairs in increasing distance order. We merge $\{a\}$ and $\{b\}$ first; this edge

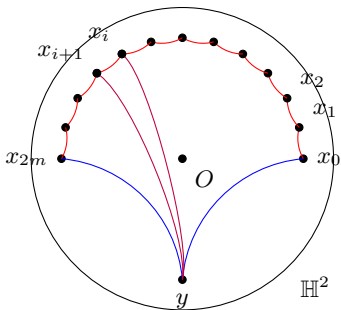

Figure 3: This figure depicts the example that proves Gromov's distortion bound is asymptotically tight. By symmetry, we can conclude that Gromov's algorithm will always return the same $\ell_\infty$ error regardless of the choice of base point $w$.

corresponds to $e_1 = (a, b)$. Then we merge $\{d\}$ and $\{e\}$ for $e_2 = (d, e)$. Next, when we explore $(e, f)$ (regardless of whether $(a, d)$ comes first or not), we merge $\{d, e\}$ and $\{f\}$ with corresponding distance 6. Then we merge $\{a, b\}$ and $\{c\}$ when we explore $(b, c)$. Finally, we merge $\{a, b, c\}$ and $\{d, e, f\}$ when both clusters are *highly connected*, and we check its corresponding fit is 8.

| $d$ | $a$ | $b$ | $c$ | $d$ | $e$ | $f$ |
|---|---|---|---|---|---|---|
| $a$ | 0 | 3 | 5 | 6 | 8 | 8 |
| $b$ | 3 | 0 | 7 | 8 | 7 | 10 |
| $c$ | 5 | 7 | 0 | 9 | 5 | 7 |
| $d$ | 6 | 8 | 9 | 0 | 4 | 5 |
| $e$ | 8 | 7 | 5 | 4 | 0 | 6 |
| $f$ | 8 | 10 | 7 | 5 | 6 | 0 |

$\Rightarrow$

| $d_U$ | $a$ | $b$ | $c$ | $d$ | $e$ | $f$ |
|---|---|---|---|---|---|---|
| $a$ | 0 | 3 | 7 | 8 | 8 | 8 |
| $b$ | 3 | 0 | 7 | 8 | 8 | 8 |
| $c$ | 7 | 7 | 0 | 8 | 8 | 8 |
| $d$ | 8 | 8 | 8 | 0 | 4 | 6 |
| $e$ | 8 | 8 | 8 | 4 | 0 | 6 |
| $f$ | 8 | 8 | 8 | 6 | 6 | 0 |

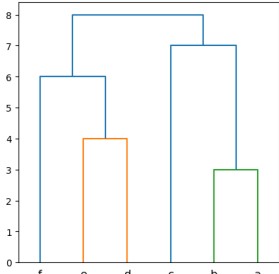

Figure 4: This figure depicts the example output $d_U$ by drawing a dendrogram.

### 6.7.2 Example for HCCRootedTreeFit

Next, given $d$, we want to find a tree fitting $d_T$ which *restricts* $r \in X$. We first compute $c_r$ and run HCCUltraFit on $d + c_r$, denote the output $d_U$. Then our tree fit $d_T$ is obtained by $d_T := d_U - c_r$, which is a tree metric. Furthermore, it restricts $r \in X$, so that $d_T(x, r) = d(x, r)$ for all $x \in X$.

| $d$ | $r$ | $a$ | $b$ | $c$ | $d$ | $e$ | $f$ |
|---|---|---|---|---|---|---|---|
| $r$ | 0 | 9 | 9 | 8 | 10 | 8 | 7 |
| $a$ | 9 | 0 | 1 | 2 | 5 | 5 | 4 |
| $b$ | 9 | 1 | 0 | 4 | 7 | 4 | 6 |
| $c$ | 8 | 2 | 4 | 0 | 7 | 1 | 2 |
| $d$ | 10 | 5 | 7 | 7 | 0 | 2 | 2 |
| $e$ | 8 | 5 | 4 | 1 | 2 | 0 | 1 |
| $f$ | 7 | 4 | 6 | 2 | 2 | 1 | 0 |

$\Rightarrow$

| $c_r$ | $r$ | $a$ | $b$ | $c$ | $d$ | $e$ | $f$ |
|---|---|---|---|---|---|---|---|
| $r$ | 0 | 11 | 11 | 12 | 10 | 12 | 13 |
| $a$ | 11 | 0 | 2 | 3 | 1 | 3 | 4 |
| $b$ | 11 | 2 | 0 | 3 | 1 | 3 | 4 |
| $c$ | 12 | 3 | 3 | 0 | 2 | 4 | 5 |
| $d$ | 10 | 1 | 1 | 2 | 0 | 2 | 3 |
| $e$ | 12 | 3 | 3 | 4 | 2 | 0 | 5 |
| $f$ | 13 | 4 | 4 | 5 | 3 | 5 | 0 |

| $d + c_r$ | $r$ | $a$ | $b$ | $c$ | $d$ | $e$ | $f$ |
|---|---|---|---|---|---|---|---|
| $r$ | 0 | 20 | 20 | 20 | 20 | 20 | 20 |
| $a$ | 20 | 0 | 3 | 5 | 6 | 8 | 8 |
| $b$ | 20 | 3 | 0 | 7 | 8 | 7 | 10 |
| $c$ | 20 | 5 | 7 | 0 | 9 | 5 | 7 |
| $d$ | 20 | 6 | 8 | 9 | 0 | 4 | 5 |
| $e$ | 20 | 8 | 7 | 5 | 4 | 0 | 6 |
| $f$ | 20 | 8 | 10 | 7 | 5 | 6 | 0 |

$\Rightarrow$

| $d_U$ | $r$ | $a$ | $b$ | $c$ | $d$ | $e$ | $f$ |
|---|---|---|---|---|---|---|---|
| $r$ | 0 | 20 | 20 | 20 | 20 | 20 | 20 |
| $a$ | 20 | 0 | 3 | 7 | 8 | 8 | 8 |
| $b$ | 20 | 3 | 0 | 7 | 8 | 8 | 8 |
| $c$ | 20 | 7 | 7 | 0 | 8 | 8 | 8 |
| $d$ | 20 | 8 | 8 | 8 | 0 | 4 | 6 |
| $e$ | 20 | 8 | 8 | 8 | 4 | 0 | 6 |
| $f$ | 20 | 8 | 8 | 8 | 6 | 6 | 0 |

$$d_T = $$

| $d$ | $r$ | $a$ | $b$ | $c$ | $d$ | $e$ | $f$ |
|---|---|---|---|---|---|---|---|
| $r$ | 0 | 9 | 9 | 8 | 10 | 8 | 7 |
| $a$ | 9 | 0 | 1 | 4 | 7 | 5 | 4 |
| $b$ | 9 | 1 | 0 | 4 | 7 | 5 | 4 |
| $c$ | 8 | 4 | 4 | 0 | 6 | 4 | 3 |
| $d$ | 10 | 7 | 7 | 6 | 0 | 2 | 3 |
| $e$ | 8 | 5 | 5 | 4 | 2 | 0 | 1 |
| $f$ | 7 | 4 | 4 | 3 | 3 | 1 | 0 |

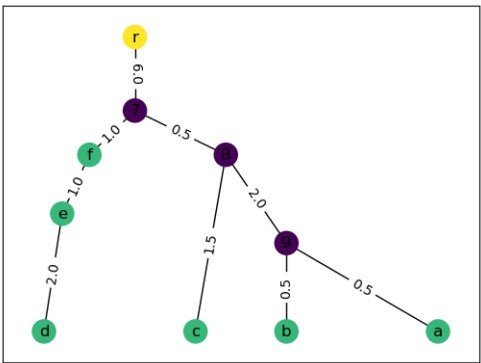

Figure 5: This figure depicts how the output $d_T$ looks like. This tree structure in fact can easily be obtained by utilizing the structure of dendrogram we computed.

## 6.8 Rooted tree structure

We discuss how the tree structure of $d_T$, the output of HCCROOTEDTREEFIT, is related to that of $d_U$. As fitting $d_U$ is an *agglomerative* clustering procedure, one may consider its linkage matrix $Z$ with specified format in scipy library (See https://docs.scipy.org/doc/scipy/reference/generated/scipy.cluster.hierarchy.linkage.html). At each step, we utilize the data to construct desired rooted tree by adding Steiner node. Detailed algorithm is as follows (For the simplicity, we will describe the procedure when $X = [n] = \{0, 1, \cdots, n - 1\}$).

## 6.9 Experiment details

### 6.9.1 TREEREP

We adapted code from TREEREP, which is available at https://github.com/rsonthal/TreeRep. We removed its dependency upon PyTorch and used the numpy library for two reasons. First, we have found that using the PyTorch library makes the implementation unnecessarily slower (especially for small input). Second, to achieve a fair comparison, we fixed the randomized seed using numpy. All edges with negative weight have been set to 0 after we have outputted the tree.

### 6.9.2 GROMOV

We used the duality of Gromov's algorithm and SLHC (observed by [13]) and simply used reduction method (from ultrametric fitting to rooted tree fitting). scipy library was used to run SLHC. This procedure only takes $O(n^2)$ time to compute.

---
**Algorithm 9** Construct rooted tree from linkage matrix $Z$
---
1: **procedure** ULTRALINKAGEFIT
2:     **Input**: distance function $d$
3:     **Output**: linkage matrix $Z$ which depicts ultrametric fit $d_U$
4: **procedure** CONSTRUCTROOTEDTREE
5:     **Input**: distance function $d$ on $\binom{[n]}{2}$ and base point $0 \leq r \leq n-1$
6:     **Output**: Rooted tree $(T, d_T)$ which fits $d$ and $d(r, x) = d_T(r, x)$ for all $0 \leq x \leq n-1$
7:     Define $m = \max_{0 \leq x \leq n-1} d(r, x)$, $c_r(x, y) = 2m - d(r, x) - d(r, y)$, and $\beta_x = 2(m - d(r, x))(0 \leq x \leq n-1)$
8:     Define $d_r(x) \leftarrow d(r, x)$ for $0 \leq w \leq n-1$
9:     $Z = \text{ULTRALINKAGEFIT}(d + c_r)$
10:    For $t \in \{0, 1, \cdots, n-2\}$:
11:       $x, y, d \leftarrow Z[t, 0], Z[t, 1], Z[t, 2]$
12:       Add node $n + t$ in $T$ with $d_r(n + t) = m - d/2$
13:       Add edge $(x, n + t)$ in $T$ with weight $d_r(x) - d_r(n + t)$
14:       Add edge $(y, n + t)$ in $T$ with weight $d_r(y) - d_r(n + t)$
15:    (Collapse node $2n - 2$ and the root node $r$)
16:    **return** $(T, d_T)$
---

### 6.9.3 NEIGHBORJOIN

For NEIGHBORJOIN, we implemented simple code. Note that it does not contain any heuristics on faster implementation. All edges with negative weight have been set to 0 after we have outputted the tree.

### 6.9.4 Hardware setup

All experiments have used shared-use computing resource. There are 4 CPU cores each with 5GiB memory. We executed all the code using a Jupyter notebook interface running Python 3.11.0 with numpy 1.24.2, scipy 1.10.0, and networkx 3.0. The operating system we used is Red Hat Enterprise Linux Server 7.9.

### 6.9.5 Common data set

For C-ELEGAN and CS PHD, we used pre-computed distance matrix from https://github.com/rsonthal/TreeRep. For CORA, AIRPORT, and DISEASE, we used https://github.com/HazyResearch/hgcn and computed the shortest-path distance matrix of its largest connected component. The supplementary material includes the data input that we utilized.

### 6.9.6 Synthetic data set

To produce random synthetic tree-like hyperbolic metric from a given tree, we do the following.

1. Pick two random vertices $v, w$ with $d_T(v, w) > 2$ (If not, then run 1 again.)
2. Add edge $(v, w)$ with weight $d_T(v, w) - 2\delta$ (for $\delta = 0.1$).
3. We repeat until new $n_e = 500$ edges have been added.
4. We compute shortest-path metric of the outputted sparse graph.

We excluded pair with $d_T(v, w) \leq 2$ because: if it is 1, then the procedure simply shrinks the edge. if it is 2, then one can consider the procedure as just adding an Steiner node (of $v, w$ and their common neighbor), which does not pose *hyperbolicity*.

### 6.9.7 Comparison

For the comparison, we have fixed the randomized seed using numpy library (from 0 to $n_{seed}$ - 1). For common data set experiments, we run $n_{seed} = 100$ times. For synthetic data set experiments, we run $n_{seed} = 50$ times.

| Error | Average $\ell_1$ error | Max $\ell_\infty$ error |
|-------|-----------------------|------------------------|
| HCC | $0.260_{\pm 0.058}$ | $1.478_{\pm 0.213}$ |
| Gromov | $0.255_{\pm 0.041}$ | $\mathbf{1.071}_{\pm 0.092}$ |
| TR | $0.282_{\pm 0.067}$ | $1.648_{\pm 0.316}$ |
| NJ | $\mathbf{0.224}$ | $1.430$ |
| QT | $1.123$ | $1.943$ |

Table 6: Experiments on unit cube in $\mathbb{R}^2$. The data set *does not* have some hyperbolicity feature so that the result kind may be different from the main experiments.

## 6.10 Detailed Comparisons

**In comparison with [1]** As [1] presented an $O(1)$ approximation that minimizes the $\ell_1$ error, it can be deduced that the total error of its output is also bounded by $O(\mathrm{AvgHyp}(d)n^3)$, while the leading coefficient is not known. It would be interesting to analyze the performance of LP based algorithms including [1] if we have some *tree-like* assumptions on input, such as $\mathrm{AvgHyp}(d)$ or the $\delta$-hyperbolicity.

**In comparison with [10]** We bounded the *total error* of the tree metric in terms of the average hyperbolicity $\mathrm{AvgHyp}(d)$ and the size of our input space $X$. As the growth function we found is $O(n^3)$, it can be seen that the *average error* bound would be $O(\mathrm{AvgHyp}(d)|X|)$, which is asymptotically tight (In other words, the dependency on $|X|$ is necessary).

While the setup [10] used is quite different, the result can be interpreted as follows: they bounded the *expectation* of the distortion in terms of the average hyperbolicity $\mathrm{AvgHyp}$ *and* the maximum bound on the Gromov product $b$ (or the diameter $D$). The specific growth function in terms of $\mathrm{AvgHyp}$ and $b$ is not known, or is hard to track. By slightly tweaking our asymptotic tightness example, it can also be seen that the dependency on $b$ should be necessary. It would also be interesting to find how tight the dependency in terms of $b$ is.

**In comparison with QUADTREE** There are a number of tree fitting algorithms in many applications, with various kinds of inputs and outputs. One notable method is QUADTREE (referred in, for example, [18]) which outputs a tree structure where each node reperesents a rectangular area of the domain. There are two major differences between QUADTREE and ours: first, QUADTREE needs to input data points (Euclidean), while ours only requires a distance matrix. Also, the main output of QUADTREE is an utilized tree data structure, while ours (and other comparisons) focus on *fitting* the metric.

We conducted a simple experiment on comparing QUADTREE and others including ours. First, we uniformly sampled 500 points in $[0,1]^2 \subset \mathbb{R}^2$ and ran QUADTREE algorithm as a baseline algorithm. While defining edge weights on the output of QUADTREE is not clear, we used $2^{-d}$ for depth $d$ edges. Note that the input we sampled is *nowhere* hyperbolic so that it may not enjoy the advantages of fitting algorithms which use such geometric assumptions.

There is a simple reason why QUADTREE behaves worse when it comes to the metric fitting problem. QUADTREE may distinguish two very close points across borders. Then its fitting distance between such points is very large; it in fact can be the diameter of the output tree, which is nearly 2 in this experiment. That kind of pairs pose a huge error.

