# OpenReview forum: "Fitting trees to $\ell_1$-hyperbolic distances"
_NeurIPS.cc/2023/Conference — NeurIPS 2023 poster_

### Official Review · Reviewer_xUok · 2023-07-06

**Soundness:** 3 good
**Presentation:** 3 good
**Contribution:** 3 good
**Rating:** 6
**Confidence:** 1

**Summary:**

The paper introduces a new algorithm, HCCRootedTreeFit, for fitting tree metrics to a given distance matrix. The algorithm is designed to minimize the ℓ1 distortion of the fit. The authors provide a detailed explanation of the algorithm, its theoretical properties, and an extensive experimental evaluation. The results show that the algorithm performs optimally when datasets are close to tree-like and when distortion is measured in the ℓ1 sense. The paper suggests that commonly used datasets, especially in geometric graph neural nets, are not well-represented by trees, indicating the need for more refined geometric notions for learning tasks with these datasets.

**Strengths:**

 The algorithm is theoretically sound, thoroughly evaluated, and performs optimally on tree-like datasets. The work provides valuable insights into dataset characteristics and has significant potential for practical applications, particularly in machine learning and data analysis.

**Weaknesses:**

The evaluation of the proposed algorithm is primarily focused on its performance on tree-like datasets. While this is certainly important, it would be beneficial to see how the algorithm performs on a broader range of datasets, particularly those that are not tree-like. This would provide a more comprehensive understanding of the algorithm's performance and its applicability to real-world problems.

**Questions:**

Please solve the weakness above

---

> ### Author Rebuttal · Authors · 2023-08-10
>
> * Thank all the reviewers for their comments and fruitful feedback. Below each review, we have posted a rebuttal that directly addresses concerns and clarifies any misunderstandings. If you wish to obtain further clarification, please reply in the relevant thread, and we will get back to you as soon as possible.
>
> * Thank you for your review and the constructive comments. We appreciate that you note our paper "provides valuable insights" into dataset characteristics. We address your comment from the "Weaknesses" section below.
>
> * The evaluation of the proposed algorithm is primarily focused on its performance on tree-like datasets. While this is certainly important, it would be beneficial to see how the algorithm performs on a broader range of datasets, particularly those that are not tree-like. This would provide a more comprehensive understanding of the algorithm's performance and its applicability to real-world problems.
>   * We understand the point that more results on a broader range of data sets would make our discussion more fruitful and comprehensive. We will conduct extra experiments on the Euclidean dataset in order to 1) compare with other popular tree fitting algorithms works on Euclidean points (such as QuadTree) as a baseline and 2) make an evaluation on a broader range of datasets, which are NOT tree-like at all.

---

> > ### Comment · Reviewer_xUok · 2023-08-18
> >
> > Thanks for the author's reply. May I ask in what application scenarios the author's method can be used and what kind of problems it can solve?

---

> > > ### Author Response · Authors · 2023-08-18
> > >
> > > There are many uses of hyperbolic embeddings in geometric graph neural nets, visualizing and representing hierarchical data, for understanding hierarchical relations amongst data points, for representing tree-like sparse graphs, and most specifically, for studying phylogenetic data (e.g., evolution of organisms, mutations of viruses, etc).

---

> > > > ### Comment · Reviewer_xUok · 2023-08-20
> > > >
> > > > Thanks for the author's response.

---

### Official Review · Reviewer_uKfW · 2023-07-07

**Soundness:** 3 good
**Presentation:** 3 good
**Contribution:** 3 good
**Rating:** 6
**Confidence:** 2

**Summary:**

The authors consider the tree fitting problem for a given distance. The authors cast the tree fitting problem as finding the relation between the hyperbolicity vector and the error of tree embedding. The authors propose an algorithmic approach with provably tight $\ell_1$ error. The authors also illustrate the advantage of the proposed approach on some “tree-like” datasets.

**Strengths:**

+ By casting the fitting tree problem as finding the relation between the hyperbolicity vector and the error of tree embedding, the authors propose provably tight $\ell_1$ error algorithmic approach.
+ The authors illustrate the advantage of the proposed approach on some “tree-like” datasets.
+ Overall, the presentation is good. (It may be better to elaborate more details on the proposed algorithmic approach in Section 3.3)

**Weaknesses:**

+ At the heart of the proposed algorithmic approach (Section 3.3), although the authors summarize their proposed approach in several Algorithms, it is hard to get the ideas how the tree is constructed. It seems the authors focus more on analysis for the proposed algorithm.
+ It seems better to include discussions about other existing approaches of tree fitting for a given distance. Besides the provable analysis of the proposed algorithmic approach, it is not clear how the proposed approach addresses some limits of existing approaches on tree fitting for a given distance.

**Questions:**

+ Could the authors elaborate the main ideas of the proposed algorithmic approach in Section 3.3? (although some algorithms are given to summarize the ideas, it is better to discuss about it, e.g., how the tree is constructed to fit a given distance, besides the analysis)?
+ Could the authors discuss how the proposed algorithmic approach improves over some simple approach, such as the QuadTree approach (which is popularly used in the context of optimal transport)?
+ If possible, please consider the QuadTree approach as a baseline for tree fitting in the experiments.
+ In Algorithm 1, could the authors explain the “highlyconnected” concept? How is it defined as in the Algorithm 1?
+ In line 269, why do the negative weights appear? (for tree metric, should the weights be nonnegative?) It seems unclear whether these approaches try to find a closest tree metric for a given distance?

Some minor points:
+ In line 58, “the table below”? which table is mentioned? It is better to have a reference for the mentioned table.
+ In Theorem 3.4, could the authors comment on the constant $C=4$? Is it a tight result or is there any chance to improve it?

---

Thank you for the rebuttal.

**Limitations:**

There are no discussions about the limitations and potential negative societal impact of their work.

---

> ### Author Rebuttal · Authors · 2023-08-10
>
> * Thank all the reviewers for their comments and fruitful feedback. Below each review, we have posted a rebuttal that directly addresses concerns and clarifies any misunderstandings. If you wish to obtain further clarification, please reply in the relevant thread, and we will get back to you as soon as possible.
>
> * Thank you for your review and the constructive comments. We appreciate that you note our paper has a good presentation. We address your comments from the "Weaknesses" and "Questions" section below.
>
> * At the heart of the proposed algorithmic approach (Section 3.3), although the authors summarize their proposed approach in several Algorithms, it is hard to get the ideas how the tree is constructed. It seems the authors focus more on analysis for the proposed algorithm.
>   * We will add more sentences about the intuition behind the algorithm in order to elucidate the idea. Our tree construction is a bottom-up approach using the sorted order of "Gromov product" over all pairs. Then we sequentially add Steiner nodes to "fit" those products.
>   * Many of the existing tree fitting methods suffer from a lack of analysis (e.g., TreeRep, NeighborJoin). Even Gromov's original work, while mathematical, isn't presented as an algorithm. It's only a few recent papers that have begun to do this analysis, including ours.
> * It seems better to include discussions about other existing approaches of tree fitting for a given distance. Besides the provable analysis of the proposed algorithmic approach, it is not clear how the proposed approach addresses some limits of existing approaches on tree fitting for a given distance.
>   * As we illustrated in the synthetic experiments, ours and other empirical algorithms behave quite differently on the synthetic data sets. This suggests that existing approaches may underperform on certain “tree-like” inputs, and that common data sets are not nearly as tree-like as researchers thing. Also, as far as we know, both the $\ell_1$ distortion bound and our approach is novel: there are some theoretical works which seek to minimize the $\ell_1$ distortion (including the recent result [1], which is actually an O(1) approximation), but they cannot be implemented practically.
> * Could the authors elaborate the main ideas of the proposed algorithmic approach in Section 3.3? (although some algorithms are given to summarize the ideas, it is better to discuss about it, e.g., how the tree is constructed to fit a given distance, besides the analysis)?
>   * The intuition behind IsHighlyConnected is that if every node in both clusters is reasonably connected with those in the other cluster, then the number of bad triangles can be favorably controlled.
> * Could the authors discuss how the proposed algorithmic approach improves over some simple approach, such as the QuadTree approach (which is popularly used in the context of optimal transport)? If possible, please consider the QuadTree approach as a baseline for tree fitting in the experiments.
>   * QuadTree needs to input data points (Euclidean), while ours only requires a distance matrix. Therefore ours works on a more generic setup. We will conduct extra experiments on the Euclidean dataset in order to 1) compare with QuadTree (and other popular) algorithms as a baseline and 2) make an evaluation on a broader range of datasets, which are NOT tree-like at all.
> * In Algorithm 1, could the authors explain the “highlyconnected” concept? How is it defined as in the Algorithm 1?
>   * The intuition behind IsHighlyConnected is that if every node in both clusters is reasonably connected with those in the other cluster, then the number of bad triangles can be favorably controlled.
>   * We will add a figure which helps to describe the concept like https://i.imgur.com/u5uyw68.jpg.
> * In line 269, why do the negative weights appear? (for tree metric, should the weights be nonnegative?) It seems unclear whether these approaches try to find a closest tree metric for a given distance?
>   * For a set of distances to satisfy a (tree) metric, all edge weights should be nonnegative. It has been shown, however, empirically that negative weights can appear in other algorithms, such as NeighborJoin or TreeRep. If the input distances satisfy a metric, then our algorithm is guaranteed to output a proper tree fitting with all non-negative weights. (This cannot be guaranteed for other tree fitting algorithms.)
> * In line 58, “the table below”? which table is mentioned? It is better to have a reference for the mentioned table.
>   * Should be labeled as "Table 1". Thanks for pointing this out!
> * In Theorem 3.4, could the authors comment on the constant $C = 4$? Is it a tight result or is there any chance to improve it?
>   * We do not think that $C = 4$ is tight but we couldn't show it. That improvement seems difficult. We also did not yet find an example which forces $C > 1$.
>   * The reason we emphasized such constant $C$ is that the usual (unweighted) CC problem can be simply solved with $C = 1$, which is actually tight. To get more detailed, if we have an unweighted graph, we are able to find a clustering that the number of disagreement edges is bounded by the number of bad triangles (with the same definition). For example, an algorithm presented at ``Aggregating Inconsistent Information: Ranking and Clustering`` (Ailon et al., 2008) achieves this bound. Also, when analyzing this problem, considering such triangles plays an important role. That is why we are somewhat optimistic that $C = 4$ can be improved with, possibly tight results on $C = 1$.

---

> > ### Comment · Area_Chair_pGit · 2023-08-18
> > **Further discussion needed?**
> >
> > Dear reviewer,
> >
> > Please let us know what you think about the rebuttal and whether you have any points you want to raise.
> >
> > Thank you,
> >
> > AC

---

> > ### Comment · Reviewer_uKfW · 2023-08-18
> >
> > Thank you for the rebuttal, I have no other raised points.

---

### Official Review · Reviewer_x4xj · 2023-07-09

**Soundness:** 3 good
**Presentation:** 2 fair
**Contribution:** 3 good
**Rating:** 5
**Confidence:** 3

**Summary:**

The paper formulate the $l_p$ tree fitting problem introduces a new algorithm, HCCROOTEDTREEFIT, for building trees in hyperbolic space by investigating the relationship between hyperbolicity (ultrametricity) vectors and the error of tree (ultrametric) embedding, which outperforms previous methods both theoretically and empirically.,



**Strengths:**

1) The authors have developed a novel approach to the tree-fitting problem that applies hyperbolic geometry and geometric group theory.
2) The developed algorithm, HCCROOTEDTREEFIT, delivers a tree metric with $l_1$ distortion bounded by polynomial function of the average hyperbolicity, while previous research delivers a tree metric with $l_{\infty}$ distortion only.
3) Comparison not only on performance but also on speed are explored, which makes a more computational sense.
4) Repeated experiments are conducted and standard deviation was analyzed.

**Weaknesses:**

The paper is not well self-contained, needing additional supplementary materials to be complete.

Table captions are not very comprehensive, e.g., Table 3 and Table 5, which increase the difficulty to understand the major experimental results.

The results lack qualitative examples to show 1) how different the proposed tree-fitting method is compared with previous tree-fitting methods. 2) how different the synthetic datasets is to real datasets.

**Questions:**

As described in the abstract, the proposed method is inspired by geometric group theory, can the authors be more specific on what is inspired the novel approach to the tree-fitting problem using which part of geometric group theory?

What does the notation in equation between line 82 and 83 mean? $\begin{pmatrix} X \\\\ 3 \end{pmatrix}$ seems to mean combinatoric number in convention but $X$ here is a set.



**Limitations:**

 The authors does not discussed about limitations, but this method is quite theoretical and is very unlikely to have potential negative societal impact

---

> ### Author Rebuttal · Authors · 2023-08-10
>
> * Thank all the reviewers for their comments and fruitful feedback. Below each review, we have posted a rebuttal that directly addresses concerns and clarifies any misunderstandings. If you wish to obtain further clarification, please reply in the relevant thread, and we will get back to you as soon as possible.
> * The paper is not well self-contained, needing additional supplementary materials to be complete.
>   * It was hard to contain every detail for the algorithms and proofs within a certain page limit so we put the most technical proofs in supplementary materials.
> * Table captions are not very comprehensive, e.g., Table 3 and Table 5, which increase the difficulty to understand the major experimental results.
>   * We will include more details on table captions; thank you for your feedback! For example, the caption of Table 3 could be “$\ell_\infty$ error (i.e.,$||d - d_T||_\infty$, max distortion) over each methods and data sets.”
> * The results lack qualitative examples to show 1) how different the proposed tree-fitting method is compared with previous tree-fitting methods. 2) how different the synthetic datasets is to real datasets.
>   * Many of the existing tree fitting methods suffer from a lack of analysis (e.g., TreeRep, NeighborJoin). Even Gromov's original work, while mathematical, isn't presented as an algorithm. It's only a few recent papers  that have begun to do this analysis.
>   * The synthetic data sets are designed to emphasize and to provide specific control on the proxy measures of  “tree-likeness”. These examples show that 1) the current widely accepted notions of hyperbolicity and tree-likeness of certain data sets may be imperfect. Also, we would argue that the quantitative analysis on such common data sets is absent as well.
> * As described in the abstract, the proposed method is inspired by geometric group theory, can the authors be more specific on what is inspired the novel approach to the tree-fitting problem using which part of geometric group theory?
>   * Gromov's original work was in Geometric Group Theory; he was a pioneer of the field. We refer to his “algorithm” which is a very small part of the original work. And $\delta$-hyperbolicity also came from his work.
> * What does the notation in equation between line 82 and 83 mean? $\binom{X}{3}$ seems to mean combinatoric number in convention but $X$ here is a set.
>   * There is also a convention for defining $\binom{X}{k}$($X$ choose $k$) for a set $X$, as the set of all $k$-element subsets of $X$ in a very similar fashion. See, for example, chapter 1.2 of Stanley’s Enumerative Combinatorics.

---

> > ### Comment · Reviewer_x4xj · 2023-08-18
> >
> > Thank you for the author's response. I would like to discuss further the last three points:
> >
> > * This question pertains to **qualitative** examples. It would be beneficial to include visualizations to demonstrate the superiority of the proposed method, especially since hyperbolic geometry is a mainstream tool in the visualization community [c1].
> > * Gromov's original work in Geometric Group Theory is cited in almost all the related works. However, my inquiry regarding **being specific** wasn't adequately addressed.
> > * In chapter 1.2 of Stanley’s "Enumerative Combinatorics" [c2] and a more recent version [c3], the convention is **NOT about subsets**. Instead, it is also used for combinatorial numbers. Could the authors clarify the origin of this convention?
> >
> > [c1] Lamping J, Rao R, Pirolli P. A focus+ context technique based on hyperbolic geometry for visualizing large hierarchies[C]//Proceedings of the SIGCHI conference on Human factors in computing systems. 1995: 401-408.
> >
> > [c2] Stanley R P. Enumerative combinatorics, v. 1[J]. Wadsworth and Brooks/Cole Mathematics Series, Monterey, California, 1986.
> >
> > [c3] Stanley R P. Enumerative Combinatorics Volume 1 second edition[J]. Cambridge studies in advanced mathematics, 2011.

---

> > > ### Author Response · Authors · 2023-08-18
> > >
> > > Strictly speaking we are not using the "group" portion of the phrase "geometric group theory." One of the origins of the field was to study the Cayley graphs of finitely generated groups. If one endows such Cayley graphs with metrics and considers these graphs as metric spaces, sometimes they are hyperbolic spaces (and the groups are referred to as hyperbolic groups). The machinery for considering both graphs and hyperbolic spaces is useful, however, despite the disconnection from a group.

---

> > > ### Author Response · Authors · 2023-08-21
> > >
> > > In chapter 1.2 of Stanley’s "Enumerative Combinatorics" (We both checked [c2] and [c3]), they stated that "Now define $\binom{S}{k}$ (sometimes denoted $S^{(k)}$ or otherwise, and read "S choose k") to be the set of all $k$-element subsets (or $k$-subsets) of $S$. And "then" they defined the usual combinatorial number. While we understand this convention is not widely accepted as the number convention, we have found that several references already used this notation, probably due to its convenience.

---

> > > > ### Comment · Reviewer_x4xj · 2023-08-21
> > > >
> > > > When talking about a **convention**, it should be widely adopted, especially in the textbook[c1,c2] mentioned in the rebuttal. In this textbook, the notation appears for combinatorial subset **only once**, while it is used for combinatorial number **more than 1000 times**.
> > > >
> > > > While I disagree with the overstatement in the rebuttal , I recognize its convenience and would stop the discussion on this minor point.

---

### Official Review · Reviewer_Fg5u · 2023-07-14

**Soundness:** 2 fair
**Presentation:** 1 poor
**Contribution:** 2 fair
**Rating:** 5
**Confidence:** 1

**Summary:**

This paper introduces a novel tree fitting algorithm named HCCRootedTreeFit. First, the authors motivate the need for a better tree fitting algorithm by stating that current methods "assume almost nothing about the underlying discrete point set, when, in fact, many real application data sets are close to hierarchical or nearly so".

Before introducing their new method, the authors introduce new proxy measures of how tree-like a dataset is. These proxy measures are used later on in several bounds, including in their main theoretical result which guarantees the existence of a tree fitting method with some nice bound on the distortion.

In their theoretical analysis, the authors mention a connection between the tree fitting problem and hierarchical correlation clustering (HCC), along with an equivalence result between tree fitting algorithms and ultrametric fitting algorithms. Then, they introduce an adapted HCC problem formulation and three algorithms named HCCTriangle, HCCUltraFit and HCCRootedTreeFit. The first algorithm solves the adapted HCC problem. The second algorithm uses HCCTriangle to solve the ultrametric fitting problem. The last algorithm uses HCCUltraFit to solve the rooted tree fitting problem.

In the experiments, the HCCRootedTreeFit algorithm is tested versus various other methods from the literature. For these experiments the authors have used both common and synthetic datasets. They observe that their method underperforms on common datasets, but performs well on synthetic datasets with respect to the $\ell_1$ norm. They attribute this observation to a lack of tree-like structure in the common datasets. They furthermore observe that their method underperforms with respect to the $\ell_\infty$ norm.

**Strengths:**

1. The paper introduces interesting new notions of hyperbolicity and ultrametricity through their hyperbolicity and ultrametricity vectors.
2. The authors derive an interesting connection between the distortion bounds for tree fitting and ultrametric fitting algorithms.
3. The new notion of hyperbolicity is used to show that, contrary to common belief, the common datasets are not very tree-like.

**Weaknesses:**

The paper is very difficult to read. Due to the very large number of definitions and technical results, the paper reads more like a collection of statements than a paper. Much of the actual content seems to be put in the appendix, adding to the feeling that the paper is not at all self-contained. Moreover, the paper appears to use many tricks from different papers without any description of these tricks. For example, in line 233, the authors refer to some paper to obtain a tree fit without actually explaining the actual procedure.

The structure of the paper also makes it rather difficult to follow, with several forward- and backward references spread throughout the paper. As an example, Subsection 3.2 appears to state several results regarding algorithms that have not yet been introduced. Then, in the next subsection, these algorithms are introduced and following these algorithms, the authors quickly throw in a proof of a result from the previous Subsection.

Another issue with the paper is that they first formulate the tree (ultrametric) fitting problem, but then, after quickly mentioning a connection to hierarchical correlation clustering, seem to actually solve an adapted version of this HCC problem. However, the introduction of their adapted HCC problem is rather unclear and uses, for example, a 'number of disagreement edges' term that is not properly defined in the paper. Also, it is not really clear to me what the connection with this problem and the original problem is.

Due to these issues I cannot adequately judge the validity of the many technical results of this paper.

There is also an issue regarding the newly proposed algorithm within the context of the experimental results. The authors show that their method underperforms on common datasets with respect to both the $\ell_1$ norm and the $\ell_\infty$ norm. Moreover, the method is quite slow compared to two of the other methods. It therefore appears that this method is only useful in synthetic settings.

**Questions:**

1. What is the exact connection between the tree fitting problem and the HCC problem?
2. How is the term $|E_t \Delta E(P_t)|$ defined precisely?

**Limitations:**

Mostly yes. They do not really address the fact that their method does not seem to have a proper use case, but they do point out its weaknesses in the experiments.

---

> ### Author Rebuttal · Authors · 2023-08-10
>
> * Thank all the reviewers for their comments and fruitful feedback. Below each review, we have posted a rebuttal that directly addresses concerns and clarifies any misunderstandings. If you wish to obtain further clarification, please reply in the relevant thread, and we will get back to you as soon as possible.
> * Thank you for the review and the constructive comments. We appreciate that you note our paper “derive an interesting connection” between the distortion bounds. We address your concerns from the “Weaknesses” and “Questions” section below.
> * The paper is very difficult to read. Due to the very large number of definitions and technical results, the paper reads more like a collection of statements than a paper. Much of the actual content seems to be put in the appendix, adding to the feeling that the paper is not at all self-contained. Moreover, the paper appears to use many tricks from different papers without any description of these tricks. For example, in line 233, the authors refer to some paper to obtain a tree fit without actually explaining the actual procedure.
>   * It was hard to contain every detail for the algorithms and proofs within a certain page limit so we put the most technical proofs in supplementary materials. However, we will absolutely work on improving its readability. For example, as you noted, line 233 -- 236 should clearly refer to Algorithm 4 in order to make the procedure more clear. Thank you for this constructive feedback!
> * The structure of the paper also makes it rather difficult to follow, with several forward- and backward references spread throughout the paper. As an example, Subsection 3.2 appears to state several results regarding algorithms that have not yet been introduced. Then, in the next subsection, these algorithms are introduced and following these algorithms, the authors quickly throw in a proof of a result from the previous Subsection.
>   * Again, the structure was not optimal as many technical proof details need to go in the Appendix due to space constraints initially. We will work on improving the structure.
> * Another issue with the paper is that they first formulate the tree (ultrametric) fitting problem, but then, after quickly mentioning a connection to hierarchical correlation clustering, seem to actually solve an adapted version of this HCC problem. However, the introduction of their adapted HCC problem is rather unclear and uses, for example, a 'number of disagreement edges' term that is not properly defined in the paper. Also, it is not really clear to me what the connection with this problem and the original problem is.
>   * There are a number of references which make a connection between the tree (ultrametric) fitting problem and the HCC problem: we addressed this question below. The connection to “adapted” HCC problem is that the quantity of `a number of disagreement edges’ (which should be properly defined using the symmetric difference notion; thanks for pointing this out) is related to $\ell_1$-notion of hyperbolicity vector, so that we made a statement using $\ell_1 / \ell_1$ tree fitting problem (Definition 2.2). This work potentially develops and broadens the notion of such known “tree-likeness”.
> * There is also an issue regarding the newly proposed algorithm within the context of the experimental results. The authors show that their method underperforms on common datasets with respect to both the $\ell_1$ norm and the $\ell_\infty$ norm. Moreover, the method is quite slow compared to two of the other methods. It therefore appears that this method is only useful in synthetic settings.
>   * We understand the point that our algorithm does not practically outperform currently known methods. As we illustrated in the synthetic experiments, we want to highlight that the current widely accepted concepts on hyperbolicity and tree-likeness of certain data sets may be imperfect, which can be observed in some synthetic examples and experiments. We suggest that future works on achieving more theoretically sound bounds will be interesting: it could also work well on common data sets?
> * What is the exact connection between the tree fitting problem and the HCC problem?
>   * The technical connection between ultrametric fitting problem and tree fitting problem can be described as the reduction from tree metric to ultrametric. In other words, if we have an adequate ultrametric fitting algorithm, then we can use it as the subroutine to develop a tree fitting algorithm. This connection was first developed by [7], which is already known. The connection between the HCC problem and the ultrametric fitting problem is also a “reduction”: there is a reduction from ultrametric to HCC. While there are a number of references describing this connection (including [2]), the connection we explored and utilized is similar to the work on [1].
>     * We included these technical details in the Appendix in order to be self-contained.
> * How is the term $|E_t \Delta E(P_t)|$ defined precisely?
>   * Given a partition $P$ over $X$, we define $E(P) := \cup_{C \in P} \binom{C}{2}$. In other words, $E(P)$ is the collection of *every* edge in the clusters of $P$. $\Delta$ denotes the symmetric difference of two sets, namely $A \Delta B := (A \setminus B) \cup (B \setminus A)$. We will add these descriptions to make the statement precise; thank you for your feedback!

---

> > ### Comment · Reviewer_Fg5u · 2023-08-14
> >
> > I would like to thank the authors for their clarifications.
> >
> > I understand that including the proofs and details of the algorithms is difficult given the page limit. However, I do strongly recommend the authors to:
> >
> > 1. add some explanations whenever results from other papers are introduced and applied;
> > 2. restructure the paper to improve readability and remove forward- and backward references when possible.

---

> > > ### Author Response · Authors · 2023-08-18
> > >
> > > Thank you! We will revise accordingly.

---

### Decision · Program_Chairs · 2023-09-21

**Decision:**

Accept (poster)

**Comment:**

This paper received 4 reviews, all above the acceptance threshold. The rebuttal period resulted in active discussions between authors and reviewers to clear up remaining concerns. The AC follows the consensus of the reviewers and suggests that the authors use the discussion outcomes to update the paper accordingly.